# RRP7A links primary microcephaly to dysfunction of ribosome biogenesis, resorption of primary cilia, and neurogenesis

Muhammad Farooq[1,2,8], Louise Lindbæk[3,8], Nicolai Krogh[1,8], Canan Doganli[1,8], Cecilie Keller[3], Maren Mönnich[1], André Brás Gonçalves[3], Srinivasan Sakthivel[1], Yuan Mang[1], Ambrin Fatima[4], Vivi Søgaard Andersen[3], Muhammad S. Hussain [5,6], Hans Eiberg[1], Lars Hansen[1], Klaus Wilbrandt Kjaer[1], Jay Gopalakrishnan[7], Lotte Bang Pedersen [3], Kjeld Møllgård [1], Henrik Nielsen [1], Shahid. M. Baig[4], Niels Tommerup[1], Søren Tvorup Christensen [3✉] & Lars Allan Larsen [1✉]

Primary microcephaly (MCPH) is characterized by reduced brain size and intellectual disability. The exact pathophysiological mechanism underlying MCPH remains to be elucidated, but dysfunction of neuronal progenitors in the developing neocortex plays a major role. We identified a homozygous missense mutation (p.W155C) in *Ribosomal RNA Processing 7 Homolog A*, *RRP7A*, segregating with MCPH in a consanguineous family with 10 affected individuals. RRP7A is highly expressed in neural stem cells in developing human forebrain, and targeted mutation of *Rrp7a* leads to defects in neurogenesis and proliferation in a mouse stem cell model. RRP7A localizes to centrosomes, cilia and nucleoli, and patient-derived fibroblasts display defects in ribosomal RNA processing, primary cilia resorption, and cell cycle progression. Analysis of zebrafish embryos supported that the patient mutation in RRP7A causes reduced brain size, impaired neurogenesis and cell proliferation, and defective ribosomal RNA processing. These findings provide novel insight into human brain development and MCPH.

[1] Department of Cellular and Molecular Medicine, University of Copenhagen, Blegdamsvej 3, DK-2200 Copenhagen, Denmark. [2] Department of Biochemistry and Biotechnology, The Islamia University of Bahawalpur, Baghdad ul Jadeed Campus, 63100 Bahawalpur, Punjab, Pakistan. [3] Department of Biology, University of Copenhagen, Universitetsparken 13, DK-2100 Copenhagen, Denmark. [4] Human Molecular Genetics Laboratory; Health Biotechnology Division, National Institute for Biotechnology and Genetic Engineering PIEAS, Jhang Road, 38000 Faisalabad, Punjab, Pakistan. [5] Institute of Biochemistry I, University of Cologne, Joseph-Stelzmann-Strasse 52, D50931 Cologne, Germany. [6] Cologne Center for Genomics and Center for Molecular Medicine Cologne, University of Cologne, Robert-Koch-Strasse 21, D50931 Cologne, Germany. [7] Institute of Human Genetics, Universitätsstrasse 1, Heinrich-Heine-University, D-40225 Düsseldorf, Germany. [8] These authors contributed equally: Muhammad Farooq, Louise Lindbæk, Nicolai Krogh, Canan Doganli. ✉email: stchristensen@bio.ku.dk; larsal@sund.ku.dk

During development of the human neocortex, neuro-epithelial cells (NECs) multiply by symmetric cell division to create an initial pool of neural progenitors (NPCs), which differentiate into apical radial glial cells (RGCs) in the ventricular (VZ) and inner subventricular (ISVZ) zones as well as to basal RGCs at the outer subventricular zone (OSVZ)[1]. RGCs undergo a complex pattern of symmetric and asymmetric cell divisions and differentiation, which expands the NPC pool and produces post-mitotic neurons[2,3]. Tight regulation of cell division and differentiation in NECs, RGCs and their progeny is important for proper NPC expansion and ultimately the number of cortical neurons[4]. Autosomal recessive primary microcephaly (MCPH; MIM #251200) is a rare neurodevelopmental disorder characterized by congenital reduction in occipitofrontal circumference (OFC) owing to hypoplasia of the cerebral cortex causing reduction in brain volume and a simplified gyral pattern. MCPH patients display reduced OFC (>3 standard deviations below the age and sex-matched populations mean) and varying degree of non-progressive cognitive dysfunction. In addition, most MCPH patients typically show sloping forehead, but no other facial or physical abnormalities[5].

MCPH is caused by mutation in one of at least 19 genes, which encode proteins essential for centrosome and cilium biogenesis and various functions in transcriptional regulation, DNA damage responses, cell cycle progression as well as cell migration, differentiation, and apoptosis[6–8]. Mutations in centrosome-associated MCPH genes affect the initial formation and later expansion of the NPC pool through mitotic failure, by premature change in cell fate from RGCs to NPCs, and by imbalanced timing of the transition from symmetric to asymmetric cell divisions of RGCs. This ultimately drains the pool of progenitors, resulting in the generation of fewer neurons and thereby a reduced size of the neocortex[9]. In this scenario, a series of centrosome-associated MCPH and MCPH-related genes, such as CPAP/CENPJ, WDR62, KIF2A, and NDE1, were shown to partly function at the level of resorption of the primary cilium;[10–13] a microtubule-based organelle[14], which during quiescence is nucleated from the centrosomal mother centriole at the cell surface to coordinate developmental signaling[15] as well as to prevent cell cycle entry and progression[16–18]. Aberrant regulation of cilium resorption is thus linked to defects in proliferation–differentiation decisions of RGCs in the developing mouse neocortex[17,19]. As an example, MCPH patient fibroblasts carrying a mutation in centrosome-associated CPAP/CENPJ present excessively long cilia, resulting in delayed ciliary disassembly and cell cycle entry, which in NPCs leads to premature differentiation and reduced brain size[11]. Despite these recent advances, little is known on the etiology underlying MCPH, which is underscored by the pleiotropy by which MCPH genes disrupt cellular processes and contribute to the disorder[20].

The current study was initiated to further our understanding of brain development and hereditary cerebral anomalies and resulted in identification of ribosomal RNA-processing protein 7 homolog A (RRP7A) as a novel MCPH gene. We demonstrate that RRP7A is highly expressed in RGCs in the developing human neocortex and the results of our in vivo and in vitro experiments in zebrafish and stem cell cultures show that mutation or depletion of RRP7A leads to reduced brain size and dysfunction in cell cycle regulation and neurogenesis. Moreover, we find that RRP7A localizes to centrosomes and the primary cilium in addition to its known nucleolar localization, and our results show that mutation or depletion of RRP7A causes ribosomal RNA-processing defects and retarded second wave of resorption of primary cilia leading to delayed S-phase entry and progression. These findings provide novel information on the mechanisms regulating brain size control and expand our knowledge on the etiology of MCPH.

## Results

**Clinical description of microcephaly patients.** A large five-generation family was ascertained from the rural area in the Rahim Yar Khan district, in the Punjab province of Pakistan. A total of 10 individuals born to consanguineous couples, six females and four males, presented with autosomal recessive microcephaly (Fig. 1a). Clinical findings along with the age and sex of the patients are shown in Supplementary Table 1. Measurements of the head circumference (HC) in eight affected individuals revealed a reduction in HC ranging from −6 to −8 standard deviations (SDs) compared with the population age and sex-related mean. We observed varying degree of intellectual disabilities and impaired cognitive function in affected family members. Four of the affected patients (V-4, V-12, V-14, and V-15) exhibited severe speech impairment. None of the affected individuals had seizures or epileptic symptoms. Cerebral MRI scans of one of the affected patients (V-14) showed decreased craniofacial ratio together with slanting forehead compatible with microcephaly (Fig. 1b). The ventricular system had normal dimensions. In addition, volume loss was seen in the corpus callosum, especially in the anterior half.

**Identification of mutation in the *RRP7A* gene.** Direct sequencing of *ASPM* (the most frequently mutated gene in MCPH cohorts from the Pakistani population) in two affected individuals did not reveal any pathogenic mutation. Mutations in *WDR62* and *CPAP/CENPJ* were excluded by linkage analysis using microsatellite markers flanking each gene. Following exclusion of the most commonly involved genes, linkage analysis was performed using Genome-Wide Human SNP 6.0 arrays. This analysis excluded linkage of rare copy number variants, and led to the identification of two possible loss of heterozygosity (LOH) regions at chromosome 2q21.3 and 22q13.1-13.2. The LOH region at chromosome 2q21.3 was excluded by short tandem repeat (STR) marker analysis. The LOH region at 22q13.1-13.2 was further confirmed and fine mapped using STR markers and single-nucleotide polymorphisms rs9306345 and rs2038062 (Supplementary Fig. 1a). Statistical analysis of haplotypes using FASTLINK software calculated a maximum LOD score $Z = 8.61$, at allele frequency ($p = 0.01$), and disease frequency of 0.001. The resultant 2.5 Mb homozygous region at chromosome 22 (40,436,371–43,001,960 bp, hg38) contained 49 annotated protein coding genes.

Owing to the large number of genes within the linkage region, we next performed whole exome sequencing (WES) using genomic DNA obtained from one affected individual. This analysis excluded involvement of rare homozygous and possibly pathogenic mutation in known MCPH genes along with any homozygous pathogenic mutation genome-wide other than the linkage region. Analysis of rare variants (allele frequency <0.01) led to the identification of a homozygous missense mutation in exon 5 of the *RRP7A* gene, which lies within the linkage region at chromosome 22. The identified mutation c.465 G > C (p. Trp155Cys) was further confirmed by direct Sanger sequencing of amplified PCR products from all of the available family members (Fig. 1c). The DNA sequencing results showed that affected family members were homozygous carriers of the mutation, while all healthy parents were heterozygous carriers, and none of the phenotypically normal individual were homozygous for the mutation (Fig. 1a). The missense mutation was not present in 300 ethnically matched controls (600 chromosomes) of Pakistani origin. The allele frequency of the mutation is 3.3E-05 in the South Asian population and the mutation is not present in 107,000 controls of other ethnicities (gnomad.broadinstitute.org). The mutation is at the fifth nucleotide from the acceptor splice

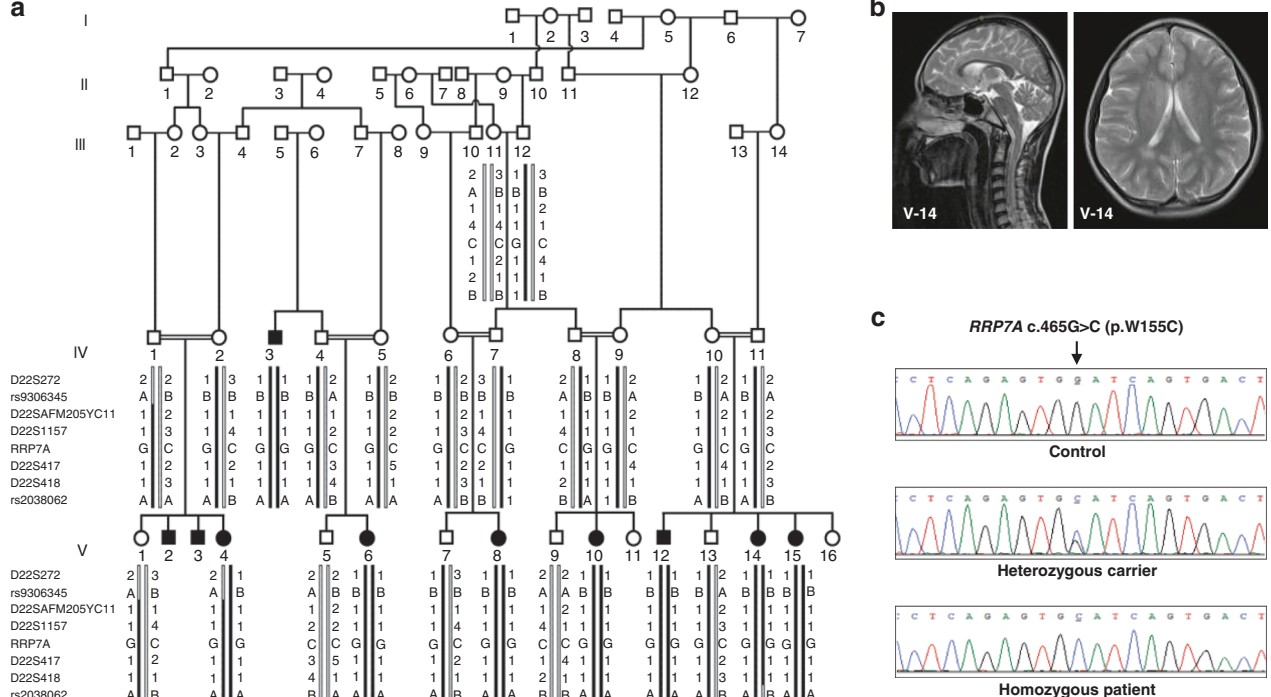

**Fig. 1 RRP7A is associated with MCPH. a** Five-generation pedigree of a consanguineous Pakistani family with ten affected individuals (black-filled symbols). STR and single-nucleotide polymorphism marker haplotypes of all analyzed individuals are shown below each symbol. Diseased haplotype is marked as filled black bar. **b** Cerebral MRI scans of patient V-14, presenting with microcephaly and slanting forehead. **c** DNA sequence of homozygous control, heterozygous carrier and homozygous patient with c.465 G > C (p.W155C) mutation in exon 5 of RRP7A.

site in exon 5. To analyze effects on RNA splicing from the mutation, we performed RT-PCR analysis of *RRP7A* mRNA using RNA obtained from a patient and a healthy control. We did not observe any effect on RNA splicing. The *RRP7A* mutation affects a tryptophan at position 155 in the protein, which is evolutionary conserved from zebrafish to humans (Supplementary Fig. 1b, c), suggesting functional importance of this amino-acid residue. In silico analysis of the effect of p.W155C mutation using the Combined Annotation Dependent Depletion (CADD) algorithm[21] resulted in a scaled CADD score of 27.1, supporting that the mutation is pathogenic.

**RRP7A is expressed in RGCs in the developing human brain.** To evaluate the spatial expression of RRP7A in the developing human brain, 3,3-diaminobenzidine (DAB) staining (bright-field immunohistochemistry) and immunofluorescence microscopy (IFM) were performed on sections of midgestation human foetal brains, in which all proliferative zones of the developing cortex are represented[22]. Examination of the parietal cortex showed prominent expression of RRP7A in RGCs in VZ and SVZ and in RG fibers in the cortical plate and marginal zone as well as localization to ependymal cilia at the VZ (Fig. 2a–k). See Supplementary Fig. 2a for indications of the individual zones stained for Vimentin that marks RGCs[23]. Evaluation of the temporal cortex and hippocampal formation revealed RRP7A localization to RGC fibers in the cortical plate (Fig. 2f–h) and at the outer surface along the fimbria (Fig. 2i–k), which is devoid of neurons at this stage as confirmed by lack of βIII-tubulin staining (Supplementary Fig. 2b). RRP7A is also expressed in other brain areas and cell types, including the meninges and endothelial cells lining the brain blood vessels (Supplementary Fig. 2c). The identification of OSVZ as the key germinal zone responsible for neocortical expansion[24] and the subsequent characterization of its population of HOPX-positive basal RGCs provided additional insight in

cortical and evolutionary neocortical expansion[25]. The prominent RRP7A immunoreactivity of apical truncated RG and, in particular, bRG following the discontinuity of the radial glia scaffold in human neocortex present at midgestation[1] indicates an important role in neural and neuronal development.

**Neurogenesis is inhibited in stem cells mutated in RRP7A.** To assess the role of RRP7A in neurogenesis, we next used the CRISPR/Cas9 system to induce deletions in exon 2 of the *Rrp7a* gene in P19CL6 cells, a mouse teratocarcinoma stem cell line that differentiates into neurons upon retinoic acid (RA) stimulation[26] (Supplementary Fig. 3a, b). Analysis of 94 clones revealed high genome editing efficiency, with an *Rrp7a* alteration in 53% of cell clones (data not shown). However, none of the observed mutations caused frameshift in both copies of *Rrp7a*. We therefore chose two clones, both compound heterozygous for a frameshift and an in-frame deletion, respectively (designated P19CL6$^{Rrp7a\Delta1/\Delta18}$ and P19CL6$^{Rrp7a\Delta8/\Delta33}$). P19CL6$^{Rrp7a\Delta1/\Delta18}$ contains one *Rrp7a* allele with 1 bp deletion and one *Rrp7a* allele with 18 bp deletion. P19CL6$^{Rrp7a\Delta8/\Delta33}$ contains one *Rrp7a* allele with 8 bp deletion and one *Rrp7a* allele with 33 bp deletion (see Supplementary Fig. 3c for detailed nomenclature). We analyzed these cell clones in parallel with two wild-type (WT) clones (P19CL6$^{Rrp7aWT\#1}$ and P19CL6$^{Rrp7aWT\#2}$), which readily differentiate into neurons (Supplementary Fig. 3d). The in-frame deletions resulted in deletion of 6 (p.53-58del) and 11 amino acids (p.49-59del), respectively, within the N-terminal domain (NTD) that makes extensive interactions with U3 small nucleolar RNA-associated protein 22 (Utp22) based on X-ray crystallography of the yeast complex[27]. Western blot (WB) analysis showed that the mutant clones express reduced levels of lower molecular mass species of RRP7A compared with the WT clones, confirming that only the *Rrp7a* alleles with in-frame deletions are translated into protein (Fig. 3a). Fluorescence microscopy and BrdU incorporation analyses showed that mutant clones

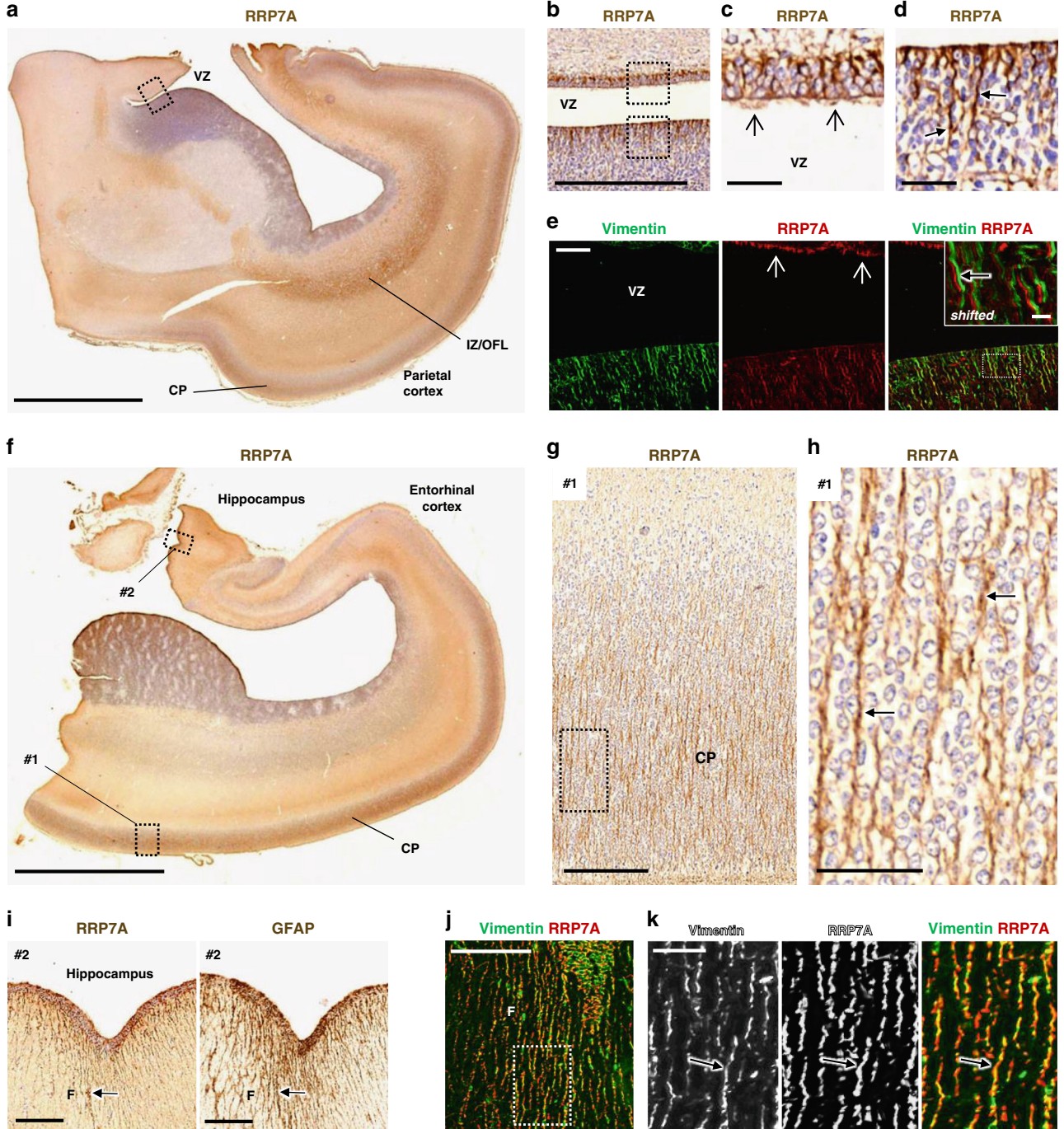

**Fig. 2 RRP7A is expressed in RGCs and cilia in the developing human neocortex aged 19 wpc. a**, **b** DAB staining depicting the expression of RRP7A in the human parietal cortex in low **a** and higher **b** magnification of the ventricular zone (VZ). Scale bars, 5 mm **a** and 0.5 mm **b**. **c**, **d** Higher magnification of zones #1 and #2 boxed in **b**. RRP7A reactivity is high in cilia (open arrows) **c** and in RGCs (closed arrows) **d**. Scale bars, 40 μm. **e** IFM analysis of the region depicted in **b** showing expression of RRP7A (red) in RGCs marked with Vimentin (green) and localization to cilia (open arrows). Scale bar, 50 μm. Insert: shifted overlay of region boxed in the merged panel showing RRP7A expression in RGCs (closed arrow). Insert scale bar, 10 μm. **f** DAB staining depicting the expression of RRP7A in RGCs in a section of the temporal cortex and hippocampal formation. Scale bar, 5 mm. **g**, **h** Higher magnifications of the zone boxed #1 in **f** showing expression of RRP7A at the cortical plate (CP). Scale bar, 0.2 mm. **h** Higher magnification of the zone boxed in **g** showing expression of RRP7A in CP RGCs (closed arrows). Scale bar, 50 μm. **i** Higher magnification of the zone boxed #2 in **f** showing expression of RRP7A (left panel) in RGCs at the outer surface along the hippocampus (F: Fimbria). RGCs (closed arrow) in this region are further marked with Glial Fibrillary Acidic Protein (GFAP) (right panel). Scale bars, 0.2 mm. Low magnification with GFAP staining is also shown in Supplementary Fig. 2a. **j** IFM analysis of the area of fimbriae shown in (i) presenting expression of RRP7A (red) in RGCs marked with Vimentin (green). Scale bar, 0.1 mm. **k** Higher magnifications of the zone boxed in **j** showing expression of RRP7A in RGCs (closed arrow). Scale bar, 25 μm. *IZ/OFL* Intermediate zone/outer fiber layer.

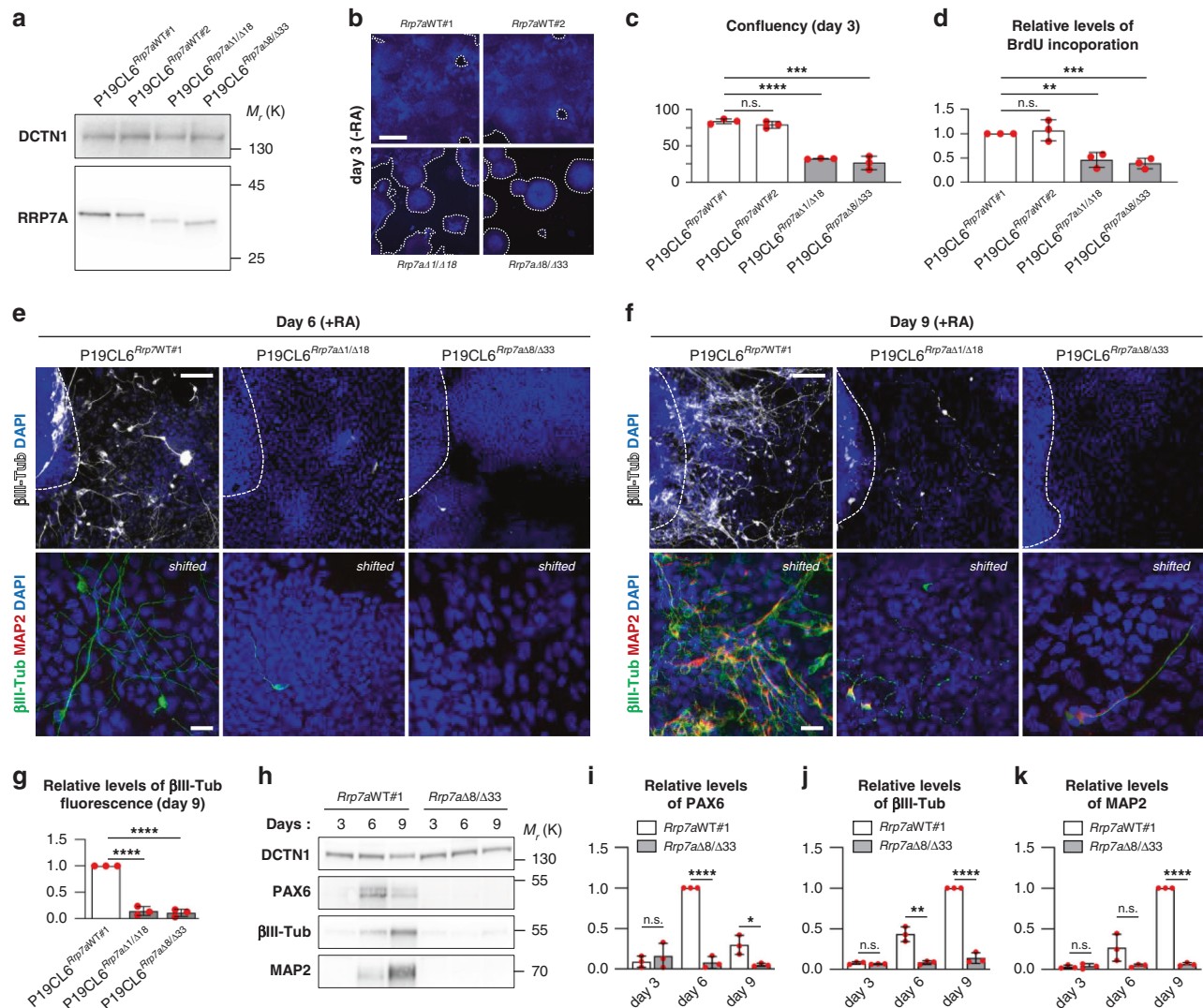

**Fig. 3 Mutation of RRP7A inhibits neurogenesis. a** WB analysis of RRP7A expression in CRISPR clones of P19CL6 mouse stem cells holding WT RRP7A (P19CL6$^{Rrp7a}$WT#1 and P19CL6$^{Rrp7a}$WT#2) or mutated RRP7A; P19CL6$^{Rrp7aΔ1/Δ18}$ (pArg59LysfsTer60/p.53-58del) and P19CL6$^{Rrp7aΔ8/Δ33}$ (p.Arg48ProfsTer15/p.49-59del). **b** IFM analysis of CRISPR clone proliferation at day 3 post seeding in the absence of retinoic acid (RA). Nuclei were stained with DAPI (blue). Dotted lines outline cell colonies. Scale bars, 1 mm. **c** Quantification of proliferation of CRISPR clones measured by cell coverage (confluency) (data average of $n = 3$ independent experiments); Rrp7aWT#2 ($P = 0.230$), Rrp7aΔ1/Δ18 ($P = 1.268E-05$), Rrp7aΔ8/Δ33 ($P = 5.372E-04$)). **d** Quantification of relative levels of BrdU incorporation in WT and mutant clones (data average of $n = 3$ independent experiments; Rrp7aWT#2 ($P = 0.617$), Rrp7aΔ1/Δ18 ($P = 0.004$), Rrp7aΔ8/Δ33 ($P = 5.635E-04$)). **e, f** IFM analysis of the ability of CRISPR clones to undergo neurogenesis after 6 **d** and 9 **e** days of RA stimulation. The stippled lines in the upper panels indicate the lining of cellular clusters from where neurons are formed in WT clones. Neurons are marked with anti-ßIII-tubulin (upper panels: white; lower panels: green) and anti-MAP2 (lower panels: red), and nuclei are stained with DAPI (blue). Lower panels are shown as shifted overlays between ßIII-tubulin and MAP2. Scale bar upper panels, 0.1 mm and lower panels, 40 μm. **g** Quantification of relative levels of βIII-tubulin fluorescence in WT and mutant clones shown in **f** (data average of $n = 3$ independent experiments; Rrp7aΔ1/Δ18 ($P = 6.244E-05$), Rrp7aΔ8/Δ33 ($P = 2.479E-04$)). **h** WB analysis of WT clone P19CL6$^{Rrp7a}$WT#1 and mutant clone P19CL6$^{Rrp7aΔ8/Δ33}$ at day 3, 6, and 9 of RA stimulation; antibodies as indicated. **i–k** Quantification of relative levels of PAX6 (day 3: $P = 0.532$; day 6: $P = 2.696E-05$; day 9: $P = 0.025$) (i), ßIII-tubulin (day 3: $P = 0.232$; day 6: $P = 0.003$; day 9: $P = 2.400E-05$) (j) and MAP2 (day 3: $P = 0.774$; day 6: $P = 0.085$; day 9: $P = 1.117E-07$) **k** shown in **h** (data average of $n = 3$ independent experiments). Data are represented as mean ± SD and significance was determined using an unpaired, two-tailed Student's $t$ test. *$P < 0.05$, **$P < 0.01$, ***$P < 0.001$, ****$P < 0.0001$, n.s.: not significant.

progress through the S-phase at a slower rate (Fig. 3b–d). To investigate whether RRP7A regulates neuronal differentiation, WT and mutant clones were cultured to the same confluency of ca. 30% followed by RA stimulation. In contrast to P19CL6$^{Rrp7a}$ WT#1, which formed neurons at linings of cell clusters at day 6 of stimulation and produced elaborate neuronal networks positive for microtubule associated protein 2 (MAP2) and βIII-tubulin at day 9, differentiation was prominently reduced in the mutant clones

(Fig. 3e–g). Impaired neurogenesis in the mutant clones was confirmed by sodium dodecyl sulfate–polyacrylamide gel electrophoresis (SDS-PAGE) and western blotting (WB) analyses, showing that P19CL6$^{Rrp7a}$WT#1 but not P19CL6$^{Rrp7aΔ8/Δ33}$ displayed significant upregulation of the neuroectodermal lineage marker Paired Box 6 (PAX6) as well as βIII-tubulin and MAP2 during RA stimulation (Fig. 3h–k). Taken together, our data suggest that RRP7A promotes cell proliferation and is required for timely neurogenesis.

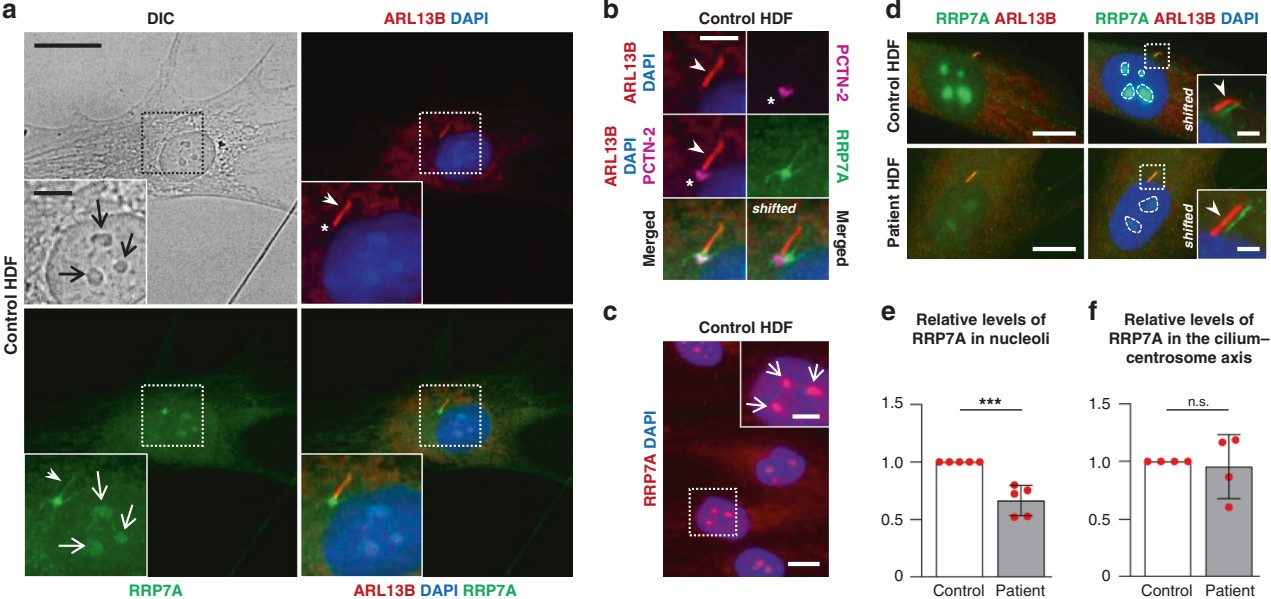

**Fig. 4 RRP7A localizes to nucleoli, centrosomes, and primary cilia. a** IFM analysis on the localization of RRP7A (green) to nucleoli (open arrows, light microscopy (LM)), the primary cilium (closed arrows, ARL13B, red) in serum-depleted cultures of control human dermal fibroblasts (HDF). The nucleus (nu) is stained with DAPI (blue). Scale bar, 20 μm. Insert scale bar, 5 μm. **b** Zoomed image of the cilium-centrosome axis depicted in **a** with the ciliary base/centrosome marked with anti-Pericentrin-2 (asterisk, PCTN-2, magenta). Scale bar, 5 μm. **c** IFM analysis of localization of RRP7A (red) to nucleoli (open arrows) in serum-supplemented cultures. Scale bar, 20 μm. Scale bar in zoomed image, 5 μm. **d** IFM analysis on the localization of RRP7A (green) to nucleoli (stippled-lined areas) and the primary cilium (ARL13B, red) in serum-depleted cultures of control HDFs (left panel) and patient HDFs (right panel). Scale bars, 10 μm. The insert shows magnification of boxed areas with shifted overlays between RRP7A (green, closed arrow) and ARL13B (red, closed arrow). The nucleus (nu) is stained with DAPI. Scale bars, 2 μm. **e** Quantification of the relative levels of RRP7A in nucleoli in control and patient HDFs shown in **d** (control = 151 cells; patient = 125 cells, data average of $n = 5$ independent experiments; $P = 0.0004$). **f** Quantification of the relative levels of RRP7A at the cilium-centrosome axis in control and patient HDFs shown in **d** (control = 84, patient = 91, data average of $n = 4$ independent experiments;; $P = 0.762$). Data are represented as mean ± SD and significance was determined using an unpaired, two-tailed Student's $t$ test. ***$P < 0.001$, n.s.: not significant.

**RRP7A localizes to nucleoli and primary cilia**. *Rrp7* was originally discovered as an essential gene in yeast encoding a protein involved in pre-rRNA processing and ribosome assembly[28] and later found to be part of the small subunit (SSU) processessome subcomplex UTP-C together with Utp22 and Casein Kinase 2[29,30]. Interestingly, this complex also binds a transcription factor (Ifh1) involved in ribosomal protein gene transcription to form the CURI complex, and in that way coordinates rRNA and protein production at different growth conditions by a titration mechanism[31,32]. A comprehensive cell-based siRNA screen in human cell lines previously implicated the human orthologue RRP7A in pre-rRNA processing[27,33]. As defective ribosome biogenesis is associated with cell cycle defects as well as neurodevelopmental disorders in combination with additional abnormalities[34,35], we investigated if the p.W155C mutation in RRP7A affects pre-rRNA processing using patient-derived (human) dermal fibroblasts (HDFs). Initially, we compared the expression of RRP7A in control and patient HDFs using WB and qRT-PCR analyses and observed reduced RRP7A protein levels, but normal mRNA levels, in the latter (Supplementary Fig. 4a, b). Addition of proteasome inhibitor MG-132 restored RRP7A levels in patient HDFs to normal (Supplementary Fig. 4a), suggesting that the p.W155C mutation leads to proteolytic degradation of RRP7A. IFM analysis of control HDFs, either cultured in the presence of serum or synchronized to growth arrest by serum depletion, showed that RRP7A is localized in the cytosol as well as to nucleoli, which is the principal site of ribosome biogenesis, as well as to the centrosome and primary cilium (Fig. 4a–c). Similar results were obtained for patient-derived HDFs except that the RRP7A signal in nucleoli was significantly reduced in these cells compared with controls (Fig. 4d–f). This phenotype was recapitulated in hTERT RPE-1 (RPE-1) cells expressing FLAG-RRP7A-GFP[WT] or FLAG-RRP7A-GFP[W155C] (Supplementary Fig. 4c–e), indicating that the p.W155C mutation impairs recruitment of RRP7A to nucleoli.

**rRNA processing and NOL6 interaction in patient cells**. To investigate rRNA processing, total RNA from control and patient HDFs was analyzed by northern blotting using oligo nucleotide probes targeting internal transcribed spacers 1 (ITS1) and 2 (ITS2). Fibroblasts from an MCPH patient with a mutation in the *WDR62* gene[36–38], which is unrelated to ribosome biogenesis, was included as a control. In human pre-rRNA processing, the 47S precursor is first cleaved at sites 01 and 02 to form 45S, followed by processing according to one of two main pathways (Supplementary Fig. 4f)[39–41]. The processing pattern in the *RRP7A* mutant differs from WT and the *WDR62* mutant in three ways (Fig. 5a). First, there is a depletion of 41S in pathway 1 (probes a and b). Second, there is an accumulation of 30S in pathway 2 (probe a). Third, there is a depletion of 21S in both pathways (probe a). These observations are consistent with a deficiency in cleavage at site A0 in formation of 18S rRNA. Accumulation of 45S or a shift towards increased use of pathway 2 could not be resolved owing to co-migration of 47S and 45S. In contrast to the deficiencies in the pathway leading to 18S rRNA, the pathway leading to 28S appeared unaffected as evidenced by equal signals corresponding to 32S and 12S, respectively. In siRNA knockdowns of RRP7A in HeLa cells, primary lung fibroblasts, and HTC-116 colon carcinoma cells, the phenotype was characterized

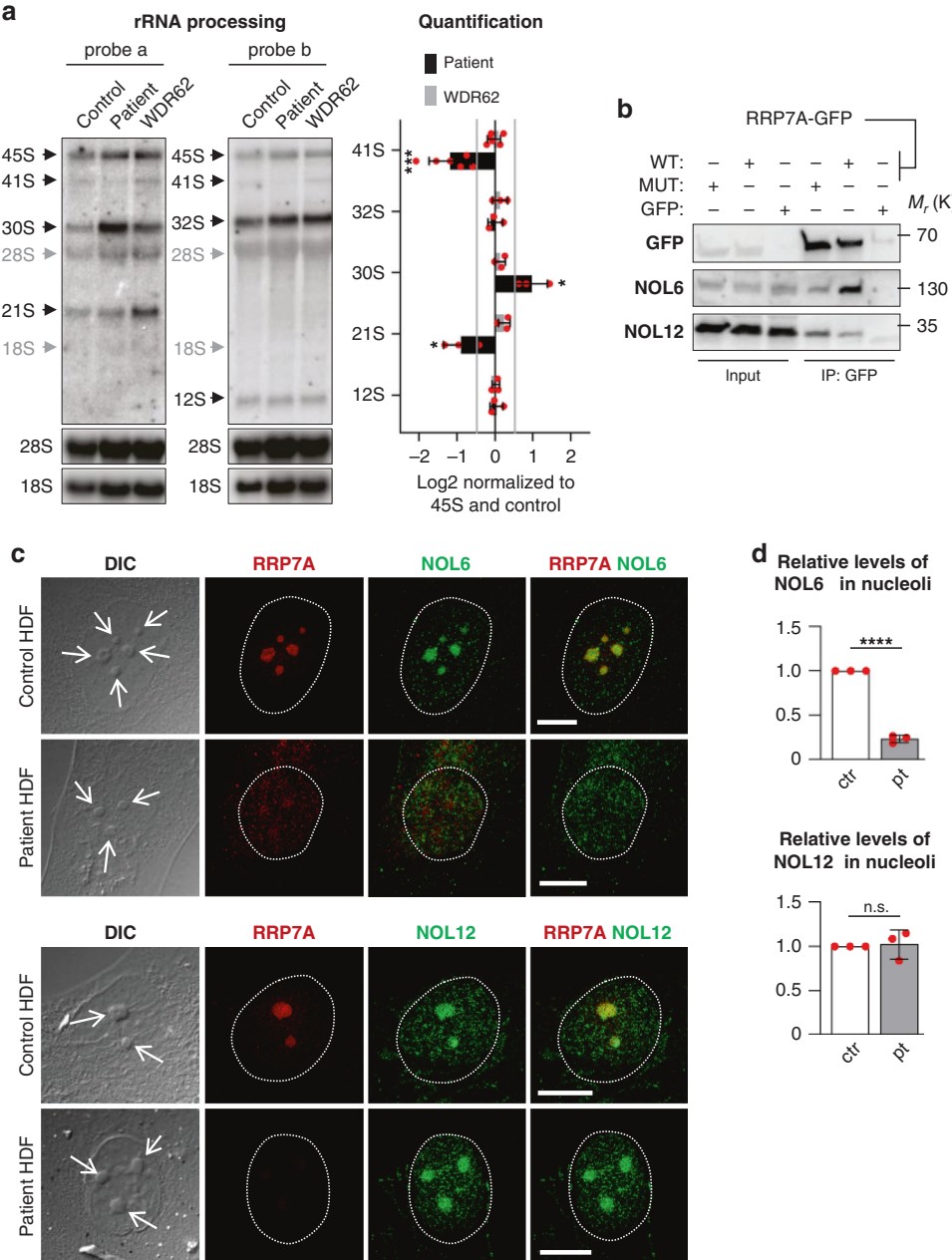

**Fig. 5 Nucleolar localization and rRNA processing are disrupted in patient HDFs. a** Left: representative northern blots of parallel gel runs of total RNA samples from control and patient HDFs with mutations in RRP7A (patient) and WDR62 (MCPH2 pt.), respectively. Black arrows indicate processing intermediates inferred from the analyses and gray arrows mark the migration of mature rRNA species as inferred from reprobing of the filters with probes targeting these RNA species. **a** Right: quantification of northern blots as log2 fold-change normalized to 45S and the control sample (data average of $n = 3$ independent experiments; 41S: $P = 0.0004$, 30S: $P = 0.017$, 21S: $P = 0.027$). Hybridization against 45S, and mature 18S and 28S rRNA were used as internal molecular markers. **b** Immunoprecipitation (IP) with anti-GFP was performed on lysates of cells transfected with constructs expressing either GFP (GFP), FLAG-RRP7A-GFP$^{WT}$ (WT) or FLAG-RRP7A-GFP$^{W155C}$ (MUT), and IPs were analyzed by WB using antibodies against NOL6 and NOL12. The experiments were repeated three times, and results from one representative experiment are shown. **c** IFM analysis on the localization of NOL6 and NOL12 (green) and RRP7A (red) to nucleoli (open arrows, differential interference contrast microscopy, DIC) in cultures of control and patient HDFs. The circumference of the nucleus is marked with a stippled line. Scale bar, 10 µm. **d** Quantification of the relative levels of NOL6 (ctr = 70 cells, pt = 66 cells, data average of $n = 3$ independent experiments; $P = 5.749E{-}06$) and NOL12 (ctr = 70 cells, pt = 73 cells, data average of $n = 3$ independent experiments; $P = 0.808$) in nucleoli in control and patient HDFs shown in **c**. Data are represented as mean ± SD and significance was determined using an unpaired, two-tailed Student's $t$ test. *$P < 0.05$, **$P < 0.01$, ****$P < 0.0001$, n.s.: not significant.

as "21S down, 18S-E down"[33]. Furthermore, re-inspection of the published northern blots[33] suggest depletion of 41S and accumulation of 30S in RPP7A-depleted cells. Thus, we conclude that the *RRP7A* mutation results in defective rRNA processing owing to a deficiency in cleavage at site A0.

The W155 residue is highly conserved and located in a patch of residues that are deeply buried owing to their interaction with Utp22 as revealed by X-ray crystallography of the Rrp7:Utp22 complex from yeast[27]. The patch is located in the linker of Rrp7 that together with the NTD forms an extensive interaction surface

with Utp22, whereas the CTD is involved in RNA-binding[27]. To investigate if the p.W155C mutation in RRP7A affects its interaction with human UTP22 (Nucleolar Protein 6; NOL6), we monitored the interaction of expressed FLAG-RRP7A-GFP[WT] and FLAG-RRP7A-GFP[W155C] with endogenous NOL6 by immunoprecipitation (IP) in HEK293T cells. The results revealed a prominent reduction in the interaction between mutant RRP7A and NOL6 (Fig. 5b), and this correlated with a reduction in nucleolar localization of NOL6 in patient HDFs (Fig. 5c, d). In contrast, RRP7A interaction with Nucleolar Protein 12 (NOL12), another pre-rRNA processing factor in the 90S processosome[42], was not compromised by the p.W155C mutation (Fig. 5b) and nucleolar localization of NOL12 was not affected in patient HDFs (Fig. 5c, d). These results suggest that defect in formation of 18S rRNA in patient HDFs is coupled to reduced interaction between RRP7A and NOL6, thereby compromising nucleolar recruitment of a fully functional RRP7A-NOL6-containing SSU processosome complex. In this context, we further evaluated the consequences of RRP7A mutation in clones of P19CL6 cells in ribosome biogenesis and nucleolar localization of NOL proteins. The results from northern blot and IFM analyses of P19CL6[Rrp7aΔ8/Δ33] show a decrease of 20S levels corresponding to human 21S compared with the WT clone, P19CL6[Rrp7aWT#1] (Supplementary Fig. 5a) as well as highly reduced nucleolar localization of NOL6 (but not NOL12) in both P19CL6[Rrp7aΔ8/Δ33] and P19CL6[Rrp7aΔ1/Δ18] compared to WT clones (Supplementary Fig. 5b–d). Thus, the P19CL6 mutants display rRNA processing and NOL6/12 localization phenotypes comparable to that of patient HDF's (Fig. 5a–d).

**Ciliary resorption and cell cycle defects in patient cells.** Similar to P19CL6 mutant clones, we observed that patient cells generally proliferated slower than controls. We therefore speculated whether the p.W155C mutation in RRP7A is associated with a delay in cell cycle entry and progression. To test this we monitored the assembly, length, and resorption of the primary cilium in control and patient fibroblasts. In cultured fibroblasts, cilium formation and resorption is induced by serum withdrawal and re-addition, respectively. Following initial serum-induced cilium disassembly at $G_0/G_1$, cells are transiently reciliated to be deciliated again during $S/G_2$ concomitant with DNA synthesis[43]. Interestingly, IFM analysis showed that RRP7A patient HDFs grown in serum-enriched medium, which normally suppresses ciliogenesis and promotes cilium resorption, display significantly higher ciliation frequency than controls (time = 0 h; Fig. 6a, b). In contrast, serum-depleted cells showed similar frequency and length of primary cilia in control and patient HDFs (Fig. 6b, c), suggesting that the RRP7A mutant HDFs fail to deciliate during conditions permissive for proliferation.

To address ciliary resorption, cells were deprived of serum for 48 h and then grown 24 h in serum-enriched medium. In accordance with previous studies[43], ciliary resorption in control HDFs proceeded in two waves, first ciliary length was reduced after 6 h of serum stimulation, followed by transient elongation at 12 h and complete ciliary resorption at 24 h post serum re-introduction (Fig. 6d–f). Patient HDFs also displayed shortened cilia at 6 h of serum stimulation, but cilia proceeded to elongate at 12–24 h post serum-introduction, which was associated with a high frequency of ciliated cells even after 24 h (Fig. 6d–f). WB analysis revealed that patient HDFs during serum re-introduction display delayed phosphorylation of retinoblastoma protein (RB) as well as CDK1 at threonine in position 161, which marks $G_1/S$ and $S/G_2$ phase transitions, respectively (Fig. 7a–c). To validate these findings, we performed a 5-bromo-2'-deoxyuridine (BrdU) incorporation assay to mark the level of S phase progression in

growth-arrested cells subjected to serum re-introduction. To this end, BrdU incorporation was evaluated in control and patient HDFs supplemented serum for 18 h, at which time point control HDFs have entered the S-phase for DNA replication[44]. This analysis revealed that S phase progression is reduced in patient cells similar to that of P19CL6 mutant clones (Fig. 7d), supporting the conclusion that the p.W155C mutation in RRP7A compromises cell cycle progression, which is coupled to the defects in the second wave of ciliary resorption.

**Mutation of *rrp7a* phenocopies MCPH in zebrafish.** Finally, to investigate the role of RRP7A in brain development in vivo, we used zebrafish as a model. Whole mount in situ hybridization (ISH) analysis of 2 days post fertilization (dpf) zebrafish embryos showed that *rrp7a* is expressed in the head of the embryo, including the brain and the eyes (Fig. 8a). We proceeded to study developmental defects associated with *rrp7a* depletion using a zebrafish line carrying a frameshift mutation in *rrp7a* (Supplementary Fig. 6a–c). Homozygous mutant progeny ($rrp7a^{-/-}$) presented with a phenotype resembling primary microcephaly, characterized by significantly reduced size of the head, but otherwise no gross abnormalities (Fig. 8b–f). Reduced head size was accompanied with significantly reduced eye size.

Reduced brain size of 3 dpf $rrp7a^{-/-}$ larvae was reflected in the transverse sections of the head where a reduction in the number of cells in the pallium was noted (Fig. 8g). In $rrp7a^{-/-}$ larvae the mutation caused premature death (7–11 dpf), whereas heterozygous mutants were viable (Supplementary Fig. 6d). Injection of wild-type fertilized oocytes with an antisense morpholino oligonucleotide (MO) designed to interfere with *rrp7a* mRNA splicing resulted in phenotypes similar to the $rrp7a^{-/-}$ mutants, supporting that the microcephaly phenotype in mutants is specific to *rrp7a* (Supplementary Fig. 6e–g). Injection with an MO designed to block the translation initiation site of *rrp7a* resulted in a slightly more-severe phenotype, possibly reflecting differences in wild-type Rrp7a protein levels within morphants and between morphants and $rrp7^{-/-}$ mutants. In $rrp7a^{-/-}$ larvae, we observed reduced expression of proliferation marker *pcna* and reduced incorporation of BrdU (Supplementary Fig. 7a), showing that S-phase progression is affected by mutation of *rrp7a*. We also observed significant reduction of some neural differentiation markers (*neurod1*, *gfap* and *isl1*) but not of others (*nes* and *elavl3*) (Supplementary Fig. 7b). Developmental markers *shha* and *fgf8a* were unaffected in $rrp7a^{-/-}$ mutants (Supplementary Fig. 7c) but results from TUNEL assays and analysis of apoptosis markers *baxa* and *bcl2* suggested increased apoptosis in $rrp7a^{-/-}$ mutants (Supplementary Fig. 8).

To test the specificity of the phenotype in $rrp7a^{-/-}$ mutants, we performed rescue experiments by injection of wild-type and mutant *rrp7a* mRNA into progeny from $rrp7^{+/-}$ zebrafish. The wild-type zebrafish *rrp7a* mRNA, but not mRNA encoding Rrp7a W152C mutant protein (Fig. 8h, Supplementary Fig. 9) partially rescued the microcephaly phenotype. Thus, the results support that the microcephaly phenotype is specific to mutation of *rrp7a* and importantly, that the phenotype–genotype relationship observed in the Pakistani family is replicated in the zebrafish model. Ultimately, to consolidate the observations from patient HDFs and P19CL6 mutant clones, we compared rRNA processing in $rrp7a^{+/+}$ and $rrp7a^{-/-}$ zebrafish (as deduced from in cross phenotypes). Northern blotting and qRT-PCR analyses showed a pronounced depletion of mature 18S rRNA in the *rrp7a* mutants compared to wild-type fish, while mature 28S levels appeared unaffected (Fig. 8i, left). In addition, we observed higher levels of processing intermediates in the $rrp7a^{+/-}$ in cross embryos displaying a microcephaly phenotype (Fig. 8i, right).

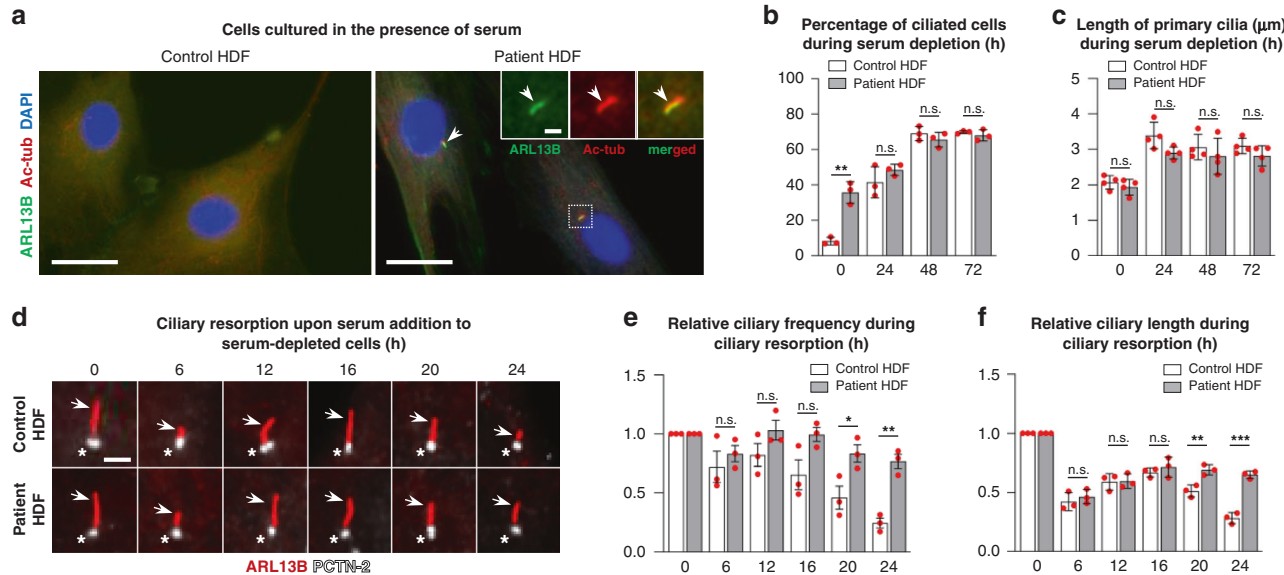

**Fig. 6 Patient HDFs display defects in resorption of primary cilia. a** IFM analysis of primary cilia in control and patient dermal fibroblasts (HDFs) cultured in the presence of serum. Primary cilia (arrows) are marked with anti-ARL13B (green) and anti-acetylated α-tubulin (Ac-tub, red), and nuclei are stained with DAPI (blue). Scale bars, 40 μm. Scale bar in zoomed image, 2 μm. **b** Quantification of ciliary frequency in control and patient HDFs serum depleted for 0–72 h (control ≥ 132 cells per time point, patient ≥125 cells per time point, data average of $n = 3$ independent experiments; 0 h: $P = 0.002$, 24 h: $P = 0.268$, 48 h: $P = 0.349$, 72 h: $P = 0.397$). **c** Quantification of ciliary length in control and patient HDFs serum depleted for 0–72 h (control ≥ 63 cells per time point, patient ≥ 82 cells per time point, data average of $n = 4$ independent experiments; 0 h: $P = 0.507$, 24 h: $P = 0.053$, 48 h: $P = 0.440$, 72 h: $P = 0.169$). **d** Overview of ciliary lengths in control HDFs (upper panels) and patient HDFs (lower panels) subjected to serum depleted for 48 h followed by serum addition for 0–24 h. Cilia are marked with anti-ARL13B (arrows, red) and centrosomes (ciliary basal bodies) are marked with anti-PCTN-2 (asterisks, white). Scale bar, 2 μm. **e** Quantification of cilia frequency based on data presented in **d** (control ≥ 131 cells per time point, patient ≥ 142 cells per time point, data average of $n = 3$ independent experiments; 6 h: $P = 0.480$, 12 h: $P = 0.176$, 16 h: $P = 0.070$, 20 h: $P = 0.039$, 24 h: $P = 0.002$). **f** Quantification of cilia length based on data presented in **d** (control ≥ 60 cells per time point; patient ≥ 60 cells per time point, data average of $n = 3$ independent experiments; 6 h: $P = 0.479$, 12 h: $P = 0.908$, 16 h: $P = 0.444$, 20 h: $P = 0.001$, 24 h: $P = 0.0004$)). Data are represented as mean ± SD and significance was determined using an unpaired, two-tailed Student's t test. *$P < 0.05$, **$P < 0.01$, ***$P < 0.001$, n.s.: not significant.

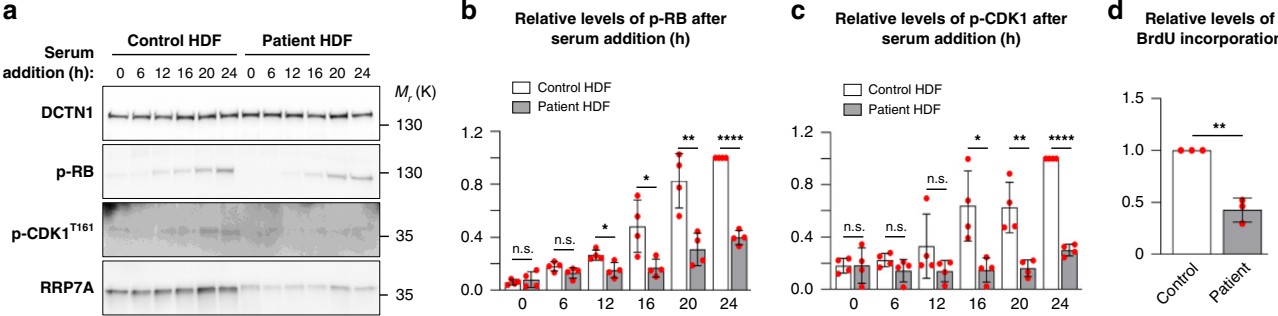

**Fig. 7 Patient HDFs display defects in cell cycle entry. a** WB analysis of the level of RRP7A expression and phosphorylation of retinoblastoma protein (p-RB) and CDK1 (p-CDK1[T161]) in control and patient HDF cells subjected to serum depleted for 48 h followed by serum addition for 0–24 h. **b** Quantification of relative levels of p-RB shown in **a** (data average of $n = 4$ independent experiments; 0 h: $P = 0.537$, 6 h: $P = 0.105$, 12 h: $P = 0.016$, 16 h: $P = 0.023$, 20 h: $P = 0.005$, 24 h: $P = 5.035E-07$). **c** Quantification of relative levels of p-CDK1[T161] shown in **a** (data average of $n = 4$ independent experiments; 0 h: $P = 0.984$, 6 h: $P = 0.157$, 12 h: $P = 0.192$, 16 h: $P = 0.014$, 20 h: $P = 0.004$, 24 h: $P = 6.358E-08$). **d** Quantification of relative levels of BrdU incorporation in growth-arrested control and patient HDFs stimulated with serum for 18 h (data average of $n = 3$ independent experiments; $P = 0.001$). Data are represented as mean ± SD and significance was determined using an unpaired, two-tailed Student's t test. *$P < 0.05$, **$P < 0.01$, ****$P < 0.0001$, n.s.: not significant.

This could be owing to increased transcription of rRNA genes to compensate for loss of 18S rRNA or a processing defect in consistence with the observation in patient fibroblasts and mouse P19CL6 cells. Zebrafish pre-rRNA processing has not been studied in sufficient detail to link the observed accumulation of intermediates to the depletion of 18S rRNA. Also, other possibilities, e.g., increased transcription of rRNA genes in the mutant, cannot be ruled out at present.

## Discussion

This study provides evidence that mutation of RRP7A causes MCPH in humans. Clinical investigation of eight homozygous carriers of a missense mutation in *RRP7A* showed severe microcephaly and mild to severe mental retardation. No other malformations or dysmorphisms were observed in the patients, and the phenotype is comparable to a clinical diagnosis of MCPH. Analysis of human foetal brain tissue sections showed

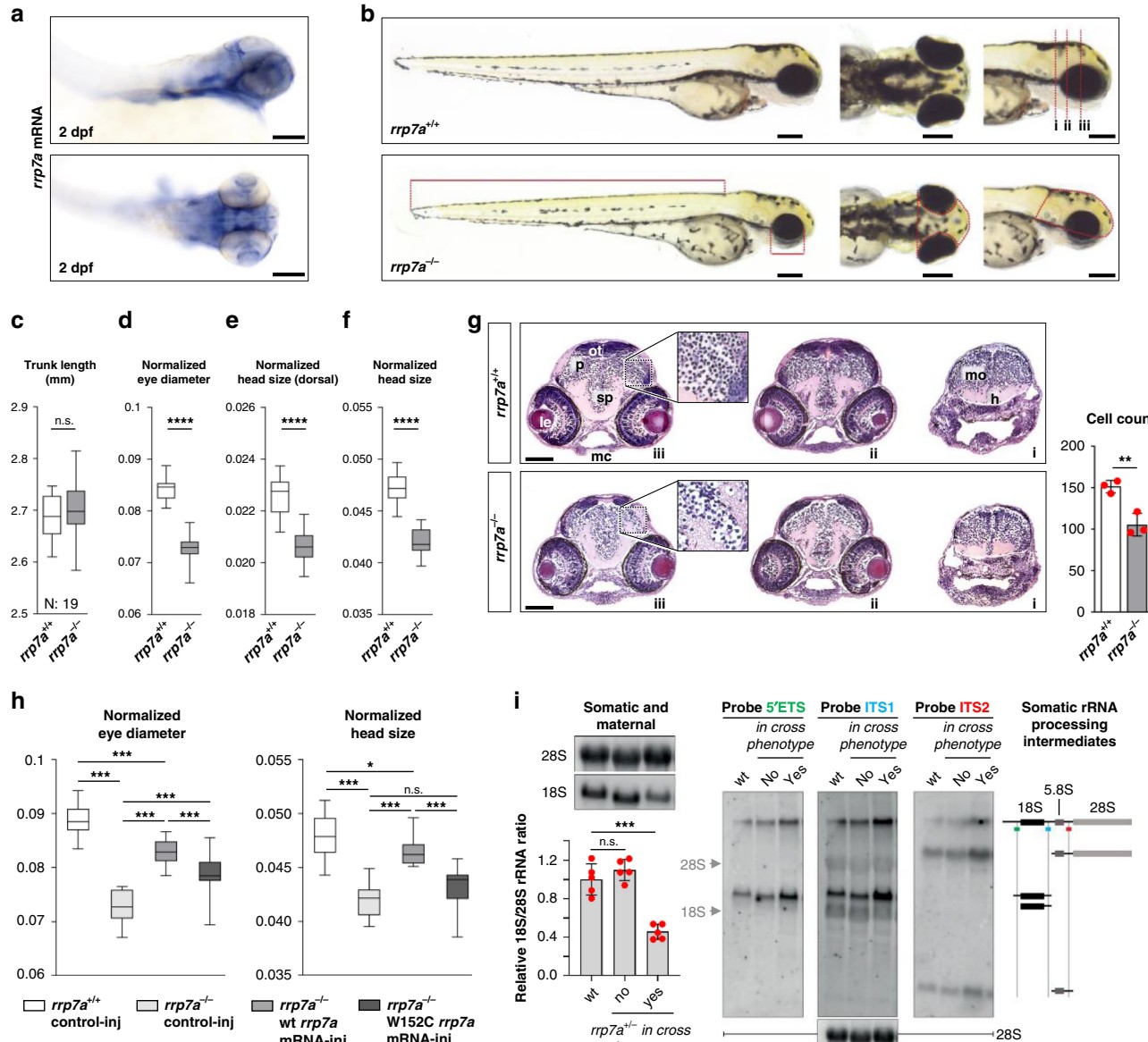

**Fig. 8 Knockout of Rrp7a in zebrafish causes brain defects resembling primary microcephaly. a** Expression of *rrp7a* in 2 dpf zebrafish analyzed by in situ hybridization. Scale bars, 0.1 mm. **b** Bright-field images showing the morphology of three dpf *rrp7+/+* and *rrp7a−/−* mutant zebrafish larvae. Scale bars, 0.2 mm. **c–f** Quantification of morphological differences between *rrp7+/+* and *rrp7a−/−* zebrafish larvae. Trunk length **c** (*P* = 0.330), normalized eye diameter **d** (*P* = 4.563E-16), dorsal head size **e** (*P* = 4.243E-10) and lateral head size **f** (*P* = 4.139E-14) were quantified (*n* = 19, biologically independent animals). Measured lengths and areas are indicated in red lines in **b** (lower panel). **g** Haematoxylin and eosin (HE) staining of transverse tissue sections from a 3 dpf *rrp7+/+* (upper panels; i–iii) and *rrp7a−/−* mutant (lower panels; i–iii) zebrafish larvae. Location of sections is indicated by red dotted lines in **b** (upper panel). Note the reduction in cells in the pallium (p) of *rrp7a−/−* compared with *rrp7+/+* (marked in zoomed box, and quantified at right, *n* = 3, biologically independent animals). Data are presented as mean ± SD (*P* = 0.007). *h* hypothalamus, *le* lens, *mc* Meckel´s cartilage, *mo* medulla oblongata, *ot* optic tectum, *sp* subpallum. Scale bars, 0.1 mm. **h** mRNA rescue of *rrp7a* mutants. Normalized eye diameter and lateral head size were quantified and plotted as box and whiskers, whiskers min to max. **i** Left: northern blot (top) of total RNA from pooled zebrafish larvae analyzed with 18S rRNA and 28S rRNA probes, respectively, and qRT-PCR (bottom) displaying relative 18S/28S rRNA ratios with wt set to 1 (data average of *n* = 5 independent experiments). Data are presented as mean ± SD, no in cross phenotype: *P* = 0.296, in cross phenotype: *P* = 1.423E-04. Right: Northern blot analysis of processing intermediates of somatic (late) rRNA in rrp7a+/− crosses and a schematic drawing of somatic rRNA processing intermediates inferred from northern blots. Hybridization against 45S, and mature 18S and 28S rRNA were used as internal molecular markers. Unpaired, two-tailed Student's *t* test. (**c–g** and **i**) and one-way ANOVA (**h**, normalized head size: box 1 vs 3 *P* = 0.035, 2 vs 4 *P* = 0.050). **P < 0.01, ***P < 0.001, ****P < 0.0001, n.s.: not significant.

strong expression of RRP7A in RGCs in the apical ventricular zone, intermediate zone, and subplate in the developing neocortex. This expression pattern is similar to the expression pattern of known MCPH genes[38,45–47] and supports a role of RRP7A in neurogenic mechanisms mediated by RGCs during cortical development. In vivo and in vitro experiments provided further

evidence for requirement of RRP7A in brain development and neurogenesis. Zebrafish embryos, homozygous for a truncating mutation in *rrp7a*, presented with significantly reduced size of the forebrain, resembling the clinical features of primary microcephaly. The normal body length of mutant embryos suggested that mutation of *rrp7a* did not cause widespread developmental

delay, but the mutation was lethal at 7–11 dpf for homozygous mutants. This observation could be due to a more severe effect of the truncating mutation, compared with the missense mutation identified in patients. The microcephaly phenotype could be rescued with injection of wild-type *rrp7a* mRNA, but injection with *rrp7a* W152C mutant mRNA failed to save the phenotype. Thus, the rescue experiments support the conclusion that the W152C mutation results in defective Rrp7a in zebrafish, similar to what we observed in humans. We further observed a reduced number of cells in the forebrain of *rrp7a* homozygous mutant embryos together with a reduced expression of neural differentiation markers, supporting that *rrp7a* deficiency affects neural development in vivo. In support of a requirement for RRP7A WT levels in neural development, we found that differentiation of P19CL6 stem cells into neurons was severely reduced in cell cultures with targeted mutations in *Rrp7a*. Furthermore, analysis of proliferation marker *pcna* and BrdU assays suggested reduced S-phase progression in zebrafish *rrp7a*$^{-/-}$ mutants. Taken together with the observed reduced rate of S-phase progression of P19CL6 *Rrp7a* mutant cells, these results indicate that RRP7A is required for multiple cellular processes, which impact on brain size control. Genome editing using CRISPR/Cas9 did not result in any cell clones homozygous for frameshift mutations in *Rrp7a* despite a high mutation rate. This could indicate negative selection against null mutations in *Rrp7a*. In addition, we observed presence of normally spliced *rrp7a* mRNA in homozygous zebrafish mutants, suggesting that the splice-site mutation is hypomorphic. These observations support the notion that *rrp7a* is an essential gene.

Analysis of patient-derived HDFs and P19CL6 *Rrp7a* mutant cells provided further information on the potential pathophysiological mechanisms underlying MCPH in the patients. RRP7A was shown to localize to nucleoli as well as to the centrosome-primary cilium axis, suggesting a function of RRP7A in close link between these organelles. Examination of RRP7A protein and mRNA levels showed that the missense mutation affects stability of RRP7A protein in patient cells, and rescue of protein level in patient cells by treatment with proteasome inhibitor supported that mutant RRP7A is partly degraded via the proteosomal pathway. We further observed significantly decreased levels of RRP7A in nucleoli of patient cells and these results were validated by experiments with RPE cells transfected with GFP-tagged RRP7A expression constructs, implying that the W155C missense mutation leads to deficiency in nucleolar recruitment of RRP7A. In line with these findings, the W155C missense mutation compromises the quality of binding with the 18S rRNA-linked SSU processome component, NOL6 and nucleolar localization of NOL6, which was unaffected in HDFs obtained from a healthy control and an MCPH patient with mutation in *WDR62*. Similarly, P19CL6 cells mutated in the site of RRP7A, which is predicted to interact with Utp22/NOL6, showed reduced nucleolar localization of NOL6. Patient HDFs, P19CL6 cells, and zebrafish mutants all displayed an rRNA processing phenotype specifically affecting production of 18S rRNA. The processing defects scaled with the genetic insults. In the cell lines, that suffered point mutations (patient HDFs) or compound heterozygous frameshift and in-frame deletion mutations (P19CL6) we observed changes of specific processing intermediate levels. In the *rrp7a*$^{-/-}$ zebrafish, the effect extended to depletion of 18S rRNA and thus an imbalance between mature rRNA species. The consistent observation of processing defects affecting 18S rRNA in the three model systems supports that defective rRNA processing is part of the pathophysiological mechanism. It will be of interest in future studies to investigate in more detail the mechanism by which mutated RRP7A becomes unstable in patient cells, and whether this is linked to reduced interaction with NOL6.

Increasing evidence points to a critical link between dysregulated ribosome biogenesis and neurodevelopmental pathologies, including microcephaly[48]. The clinical features observed in patients mutated in RRP7A, however, differ from the usual findings in patients with mutations in genes associated with ribosome biogenesis. For example, Diamond-Blackfan anemia (MIM # 105650) is characterized by macrocytic anemia, short stature, and variable malformations of craniofacial skeleton, eyes, heart, limbs, and digits[49]. Treacher Collins syndrome (MIM # 154500) is characterized by variable craniofacial malformations and Shwachman Diamond syndrome (MIM # 260400) is characterized by dysfunction of the bone marrow and pancreas, growth retardation, and skeletal abnormalities[49,50]. Nevertheless, microcephaly, seizures, growth retardation, hypotonia, and genitourinary abnormalities were reported in patients carrying missense mutations within the gene encoding the 60S ribosomal protein L10 (*RPL10*)[51,52] and mild microcephaly, hearing loss, and dysmorphism was reported in patients with de novo mutations in *RPS23* encoding ribosomal protein uS12[53]. These reports suggest that defective ribosome biogenesis may lead to microcephaly in combination with additional abnormalities. Imbalance in mature ribosomal RNA stoichiometry resulting from depletion of 18S rRNA may cause nucleolar stress, deregulated cell cycle progression, and apoptosis and thus account for the microcephaly in patients. This is reminiscent of the pre-rRNA processing defect and the morphological phenotype of the zebrafish *rrp7a*$^{-/-}$ mutants. However, even a milder kinetic effect on ribosome biogenesis, as observed in the cell line models, may impact ribosome production and cell growth. Indeed, the patient HDFs grow considerable slower that WT cells. Furthermore, evidence is accumulating that changes in ribosome dose can dramatically affect the cellular proteome and cell fate[54,55]. Thus, in addition to similar effects on rRNA processing, the growth phenotypes in the three models appear related, with slow growth in HDFs, defects in neuronal differentiation in P19CL6 stem cells and reduced brain size in zebrafish. Taken together, the three model systems that were subjected to different genetic insults of rrp7a show corresponding defects in rRNA processing and growth and differentiation phenotypes supporting the interpretation that one consequence of the p.W155C patient mutation is on rRNA production, although further work will be required to delineate the precise consequences of ribosome insufficiency on brain development such as in proliferation–differentiation decisions of RGCs in the developing neocortex.

A defect in the timely resorption of primary cilia suppresses cell cycle entry and progression and is an emerging hallmark of primary microcephaly caused by mutations in centrosome-associated MCPH genes[10–13,56]. Analyses of ciliary resorption and cell cycle control in growth-arrested HDFs revealed that patient cells have defects in the second wave of deciliation in concurrence with delayed G$_1$/S and S/G$_2$ phase transitions and decreased rate of S phase progression. These findings support a functional role of RRP7A in cell cycle control in the context of the primary cilium, albeit in contrast to the function of other MCPH proteins such as CPAP/CENPJ, WDR62, and KIF2A, in which mutations mainly disrupt the first wave of resorption[13,56]. Ciliary disassembly is regulated by distinct sets of signaling events that control destabilization and depolymerization of axonemal microtubules[18,57], suggesting that the RRP7A W155C missense mutation is associated with imbalanced regulation of events that distinctively control the second wave of resorption. Further, our observation that patient HDFs display aberrant formation of primary cilia under conditions permissive for cell proliferation, favors the conclusion that RRP7A both safeguards against aberrant ciliary assembly in cycling cells, and ensures timely ciliary resorption. Both of these events are characteristic of ablation of,

e.g., the microtubule depolymerizing kinesin, KIF24, which is activated by ciliary base recruitment of the NIMA Related Kinase 2 to promote ciliary disassembly and to negatively regulate ciliary assembly through its interaction with Centrosomal Protein of 110 kDa (CP110)[58,59]. Future studies should therefore aim to delineate the mechanisms by which RRP7A mutations compromise regulatory events in ciliary formation and resorption, and how those mechanisms contribute to the regulation of neurogenic processes during brain development.

In conclusion, we present the identification of *RRP7A* as a novel disease gene in MCPH. Analyses of human foetal brain sections, zebrafish, cell models, and primary cells from patients show that RRP7A levels and functionality is critical in neocortex development and neurogenesis and provide evidence for defective ribosome biogenesis as well as resorption of the primary cilium as pathophysiological mechanisms. Interestingly, recent studies have shown that microcephaly caused by neural tissue infection of flaviviruses, including Zika virus (ZIKV) as well as Japanese encephalitis, West Nile, and Dengue viruses, in part is associated with localization of virus-encoded proteins to both nucleoli[48,60] and the base of primary cilia[61] in conjunction with dysregulated ribosome biogenesis[48], aberrant length control of primary cilia, and uncontrolled cell fate determination of NPCs[61,62]. In support of both nucleolar and ciliary functions, co-IP analyses and global interactomics confirmed ZIKV-host interactions to key regulatory components of both the cilium-centrosome axis and nucleoli, including NOL6[61,63]. To our knowledge, *RRP7A* presents the first evidence of an MCPH gene involved in the borderline between nucleoli and primary cilia. Further investigations are warranted to delineate the specific role of RRP7A at the cilium-centrosome axis in cell cultures as well as in the developing human brain and how mutation in RRP7A compromises ciliary functions, including ciliary resorption and developmental signaling in cell cycle control and cell fate decisions.

## Methods

**Patient samples.** The study followed the declaration of Helsinki and was approved by the Institutional Research Ethics Committee, National Institute for Biotechnology and Genetic Engineering, Faisalabad, Pakistan. The five-generation family originates from the rural area in the Rahim Yar Khan district, in Punjab province of Pakistan. Informed consent was obtained from all participating individuals or their parents for the collection of blood and/or skin biopsy samples, genetic analyses, and publication of photos and genetic information. Blood samples from all available family members were collected in separate appropriate tubes for genomic DNA/RNA isolation. Skin biopsies from affected individuals were collected in Dulbecco's modified Eagle's medium (DMEM) for cell cultures. Genomic DNA was isolated from peripheral blood cells using phenol-chloroform extraction.

**Magnetic resonance imaging.** Magnetic resonance imaging (MRI) scan was performed for one affected individual at the Sheikh Zayed Hospital, Rahim Yar Khan. Multiplanar, multisequential FLAIR and diffusion images of brain MRI were obtained in axial, coronal and sagittal views and images were analyzed using *syngo* fastView (version 1.0.0.3, Siemens AG).

**Genome-wide SNP array mapping and linkage analysis.** To identify regions of homozygosity, deletions, or duplications, samples were genotyped on Genome-Wide Human SNP array 6.0 (Affymetrix Inc., Santa Clara, CA, USA). The analysis was performed in four patients (one from each loop) according to the manufacturer's standard protocol (Affymetrix). Data analysis was performed using Chromosomal Analysis Suite (ChAS) and FASTLINK software tools[64]. Polymorphic STR markers (D22S272, D22SAFM205YC11, D22S1157, D22S417, D22S418, primers are available upon request) were used to genotype all available family members to identify and map the linked chromosomal region. STR markers were PCR amplified and alleles were separated in an ABI 3130xl Genetic Analyzer (Applied Biosystems, Foster City, CA, USA) according to the manufacturer's instructions. Results were analyzed using Peak Scanner Software, version 1.0 (Thermo Scientific, Waltham, MA USA).

**Genetic analysis.** Direct sequencing of *ASPM* and linkage analysis of *WDR62* and *CENPJ*, using microsatellite markers flanking each gene, was used to exclude the most common MCPH genes in the Pakistani population. Following exclusion of

the most commonly involved genes, linkage analysis was performed using FASTLINK software. WES was performed on genomic DNA extracted from an affected individual. Capture was performed using the Nextera Rapid Capture Exome Enrichment Kit and sequencing was done using a paired-end $2 \times 100$ bp protocol using an Illumina Hi-seq 2000 system (Illumina, San Diego, CA, USA). Data were filtered and aligned to the Human Genome reference (hg38) using Burrows Wheel Alignment. Variants were analyzed using the Genome Analysis Toolkit. Variants with minor allele frequency of <0.01 were filtered against known databases of genomic variants (dbSNP138, ExAC). Of the remaining variants, only rare variants affecting protein structure and predicted deleterious by at least one of the prediction algorithms SIFT and Polyphen2 were considered. The *RRP7A* mutation c.465 G > C was confirmed by direct Sanger sequencing of a PCR-amplified exon-fragment in all available samples from patients and unaffected heterozygous parents. The following primers were used: 5′-GACTACATGTGTCACCCAGC-3′ (forward) and 5′-GGATCTCCTTAGCACCTGTG-3′ (reverse). Sequencing was performed in an ABI 3130xl genetic analyzer using the ABI Prism Big Dye Terminator v3.1 Cycle Sequencing Kit (Applied Biosystems) according to the manufacturer's protocol. Sequences were analyzed using chromaspro software (chromaspro 2.1, Technelysium Pty Ltd, South Brisbane, Australia). The c.465 G > C mutation leads to the introduction of an HpyCH4V restriction enzyme site in the mutant. Analysis of the mutation in DNA samples obtained from 300 ethnically matched Pakistani control individuals were performed by restriction digestion on amplified PCR products, using the HpyCH4V restriction enzyme. HpyCH4V restriction enzyme digestion products were analyzed on 1.5% agarose gels.

**Multiple alignment of RRP7A.** Multiple protein sequences alignments to determine the conservation of the mutated amino acid (p.W155C) was done using the online alignment tool (http://www.uniprot.org/). RRP7A ( > NP_056518.2) has 23% sequence similarity with the Ribosomal RNA-processing protein 7 [(RRP7) *S. cerevisiae*].

**Total RNA isolation, RT-PCR and qRT-PCR.** To check for RRP7A splicing errors, total RNA was isolated from whole blood obtained from a patient (V.14) and a healthy control individual. Blood for RNA was collected in PAXgene blood RNA tubes, and was isolated using a PAXgene® Blood RNA Kit (PreAnalyticX). One microgam of total RNA was reverse transcribed using SuperScript™ II (Invitrogen, Carlsbad, CA), and oligo-dT primers (Invitrogen). To detect aberrant splicing, full-length *RRP7A* gene products were PCR-amplified from first strand cDNA with gene specific primers 5′-ATGGTGGCGCGCAGGAGGAA-3′ (forward) and 5′-GTACGGTCGGAATTTGCG-3′ (reverse). The PCR products were analyzed on 1% agarose gels, and subsequently directly sequenced in both directions. To check for the analysis of ribosomal RNA-processing defects, total RNA was extracted from fresh cultures of primary fibroblasts obtained from an affected individual (V.14), an MCPH patient homozygous for *WDR62* mutation (loss of function mutation due to microduplication (unpublished data)) and a normal control using QIAzol® lysis reagent (Qiagen Inc., Valencia, CA) in accordance with the manufacturer's guidelines. Quality of extracted RNAs were checked by running samples in 1% agarose gels and quantification was done using NanoDrop™ ND-1000 spectrophotometer (Thermo Fisher Scientific Inc., Wilmington, USA). In order to assess expression of *RRP7A* mRNA in primary fibroblast cultures, qRT-PCR was performed using Brilliant III Ultra-Fast SYBR® Green QPCR master mix (Agilent Technologies). The qRT-PCR reactions were performed in triplicate and were analyzed on an ABI 7500 fast system. The data was normalized against the average expression value of two housekeeping genes (*GAPDH* and *B2M*, primer sequences available upon request).

**Human brain tissue samples and immunohistochemistry.** Eight brains from 7 weeks post conception (wpc) to 21 wpc were analyzed. Nineteen wpc mid-gestation foetuses were chosen for the entire immunohistochemical analysis as material from all brain regions was available and all cell types present at this stage. The material was obtained from legal abortions. Written and oral informed consent was given according to the Helsinki declaration II, approved by the Research Ethics Committee of the Capital Region (KF–V.100.1735/90). Immediately following operation, the samples were dissected into blocks and fixed for 12–24 h at 4°C in either 10% neutral buffered formalin, 4% Formol-Calcium, Lillie's or Bouin's fixatives. The specimens were dehydrated with graded alcohols, cleared in xylene, and paraffin embedded. Serial sections, 3–10 µm thick, were cut in transverse, sagittal, or horizontal planes, placed on salinized glass slides, and used for immunohistochemical experiments[65]. Immunohistochemical analyses were performed on deparaffinised sections of human brain 19 wpc. Samples were carefully treated in heat-induced antigen retrieval using citrate buffer pH 6.0, prior to blocking of unspecific binding and incubation with primary antibodies overnight[65]. Immunofluorescence staining was prepared as for bright-field microscopy. Primary antibodies were mixed in proper dilutions, and tissue was incubated overnight. Subsequently, sections were washed in TBS and incubated for 45 min at RT in Alexa Flour antibodies of the desired wavelengths. Antibody dilutions and manufacturers are listed in Supplementary Tables 2 and 3. Sections were mounted using Shandon™ ImmuMount (Thermo Fisher Scientific).

**Generation of CRISPR clones of P19CL6 cells**. CRISPR was performed as previously described[66]. In brief, 23 nt oligos targeting the second exon of *Rrp7a* were cloned into a pSpCas9(BB)-2A-GFP (PX-458) vector, amplified in *Eschericia coli* DH10α, and purified using NucleoBond Xtra Midi EF Kit (Machery-Nagel). Plasmid inserts were sequenced at Eurofins (MWG Operon, Ebersberg, Germany). P19CL6 cells (Riken Cell Bank, Ibaraki, Japan) were transfected using the Nucleofector device II (Amaxa, Biosystems Lonza, Switzerland) and Nucleofector Kit V (Amaxa Biosystems Lonza, Switzerland)[67]. Clones were evaluated by Indel Detection by Amplicon Analysis in which amplicons were labeled and DNA fragments were analyzed by automated capillary electrophoresis for detection of indels induced by CRISPR/Cas9 mediated gene targeting of Rrp7a, as described[68]. In brief, tagged forward primer (5′-AGCTGACCGGCAGCAAAATTGCCCGTG GATGGGTGGATG-3′) was used for amplification of the DNA fragment covering the sgRNA-binding sites, and analyzed using the ABI 3010 sequenator (ABI/Life Technologies, USA) under conditions specified by the manufacturer. Raw data was analyzed using the Peak Scanner Software V1.0 (ABI/Life Technologies, USA). Clones were sequenced by Sanger sequencing using forward primer 5′-CCCGTG GATGGGTGGATG-3′ and reverse 5′-CACCAGGATTCTGTAGGGC-3′ and named according to the total number of nucleotides deleted. Two guides were used to generate clone P19CL6$^{Rrp7aΔ8/Δ33}$: Sg1 5′-AGACAGCACCGAGTTCGCCAA GG-3′ and Sg2 5′-CCCGATCGAACCCTTTTTATCCT-3′. Only Sg2 were used for clone P19CL6$^{Rrp7aΔ1/Δ18}$. Two wild-type clones (P19CL6$^{Rrp7a\ WT\#1}$ and P19CL6$^{Rrp7a\ WT\#2}$) were chosen as control cell lines, and their ability to differentiate was confirmed upon 9 days of RA treatment, in which both cell lines readily form elaborate networks of βIII-tubulin positive neurons (Supplementary Fig. 3d).

**Culturing of HDFs, hTERT RPE-1, and HEK293T cells**. Skin punch biopsies were obtained from an affected individual homozygous for *RRP7A* c.465 G > C and from a MCPH patient homozygous for a *WDR62* mutation. The samples were treated with Dispase II (1.5 U ml⁻¹, Roche) in phosphate-buffered saline (PBS) at 4°C for 3 h to separate epidermis from dermis. The dermis was then incubated at 37°C and 5% CO₂ in DMEM (Invitrogen) supplemented with 10% foetal bovine seuym (FBS, Sigma; F9665), 1% GlutaMAX™ (GIBCO®), 100 IU ml⁻¹ penicillin (pen), and 100 µg ml⁻¹ streptomycin (strep) (Sigma; P0781). Culture medium was changed after every 3−4 days and a pool of human dermal fibroblasts (HDFs) was expanded by further sub-culturing. Similarly, control HDFs were obtained using skin biopsy from 34 years old healthy control female of Asian origin. hTERT RPE-1 (RPE-1; ATCC, CRL-4000) cells were cultured in DMEM medium supplemented with 10% FBS and 5% pen/strep. HEK293T cells (ATCC, CRL-3216) were cultured in DMEM containing high ʟ-glucose concentration (GIBCO®) supplemented with 10% FBS and 1% pen/strep. Culture medium was changed every 2–3 days for both RPE-1 and HEK293T cells.

**Constructs and transfection**. Using first strand cDNA obtained from a healthy donor as templates, the full-length coding sequence of *RRP7A* (GenBank accession number NM_015703.4) was PCR amplified and cloned into the mammalian expression vector pEGFP-n1 using *Eco*RI and *Bam*HI restriction sites. Two full-length wild-type expression constructs for RRP7A were generated; one containing a green fluorescent protein (GFP)-tag at the *C*-terminus and another containing a FLAG-tag at the *N*-terminus and a *C*-terminal GFP-tag, using pEGFP plasmid (Addgene). The mutation p.W155C was introduced into these vectors using QuikChange II Site-Directed Mutagenesis Kit (Agilent technologies Inc.). The primers used were 5′-GGCATTCACAAGTGCATCAGTGACTACGC-3′ (sense) and 5′-GCGTAGTCACTGATGCACTTGTGAATGCC-3′ (anti-sense). Expression constructs were verified by Sanger sequencing, and expressed in RPE-1 and HEK293T cells. Cells were seeded in DMEM without pen/strep, and transfected using FuGENE6 HD Transfection Reagent (Promega) when they reached ~60% confluence[69]. As a control in IP analyses, HEK293T cells were also transfected with the empty vector pEGFP-C1.

**Immunocytochemistry**. Cells were fixed in 4% paraformaldehyde (PFA) and permeabilised in 0.2% Triton X-100. Blocking was performed using 5% bovine serum albumin (BSA), and cells were incubated in primary antibodies for 1.5 h at RT or overnight at 4°C. Secondary antibodies were incubated for 45 min, and 4′,6-diamidino-2-phenylindole (DAPI) staining performed for 30 seconds prior to mounting in mounting media containing 2% *N*-propyl gallate and sealed with nail polish[70]. Details on antibodies concentrations and manufacturers are listed in Supplementary Tables 2 and 3. Fluorescence and differential interference contrast (DIC) images were captured on a fully motorized Olympus BX63 upright microscope with an Olympus DP72 color, 12.8-megapixel, 4140 × 3096-resolution camera and with a fully motorized and automated Olympus IX83 inverted microscope with a Hamamatsu ORCA-Flash 4.0 camera (C11440-22CU). The software used was Olympus CellSens Dimension version 1.7, which was able to do deconvolution on captured z stacks, and images were processed for publication using Image J version 2.0 and Adobe Photoshop CS6 version 13.0. Quantifications of immunofluorescence intensities and cell confluency were performed in Cell Sense (Olympus). Careful outline of the cilium-centrosome axis as well as nucleoli were drawn using the outline tool, and mean intensities were obtained. ARL13B

and Acetylated α-tubulin (Ac-tub) were used as ciliary markers to study the formation, length and disassembly of primary cilia. All data were gathered for $n ≥ 3$.

**SDS-PAGE and WB analyses**. After washing in 1× ice cold PBS, cells were lysed in 1× SDS lysis buffer with Complete, ethylenediaminetetraacetic acid (EDTA)-free Protease Inhibitor Cocktail (04693132001, Roche, Basel, Switzerland), with/without 2 mM sodium orthovanadate (S6508, Sigma). Cells were sonicated thrice for 5 seconds on ice (30 seconds for P19CL6 clones) and centrifuged at 20,000 × *g* for 20 min at 4°C (10°C for P19CL6 clones). Cleared supernatants were collected, and protein quantification was done using DC™ protein assay kit (Bio-Rad laboratories Inc. Hercules, CA, USA) according to the manufacturer's instructions. Then, samples were denatured in NuPAGE LDS sample buffer (NP0007, Invitrogen) at 95°C for 5 min, separated on gels (NuPAGE, Invitrogen) under reducing conditions and blotted onto nitrocellulose membrane (LC2006, Invitrogen). Membranes were blocked in 5% skim milk or BSA in TBS with 0.1 % tween20 (TBST) buffer followed by incubation with primary antibodies overnight and horseradish peroxidase-conjugated secondary antibodies for 2 h at RT. For reprobing, membranes were washed thrice in TBST with shaking at RT and then incubated with striping buffer (46430, ThermoFisher Scientific) for 20 min at RT. Membranes were then washed in TBST and blocked as described above. Membranes were developed using the FUSION FX chemiluminescence system (Vilber Lourmat) or on x-ray film, and processed for presentations using Adobe photoshop CS6 version 13.0. Band intensities were analyzed using ImageJ version 2.0 and UN-SCAN-IT gel version 6.1, and statistical calculations were performed on $n = 3$ or more. Details on antibodies concentrations and manufacturers are listed in Supplementary Tables 2 (primary antibodies) and 3 (secondary antibodies).

**Co-IP**. HEK293T cells were transfected the day before IP. Cells were harvested in ice cold, modified EBC buffer (120 mM NaCl, 50 mM Tris-HCl, 0.5% NP-40 and 5 mM EDTA) and cell extracts were incubated for 15 min on ice and cleared by centrifugation at 20,000 × *g* at 4°C for 15 min. In total, 10% of the cleared cell extracts (input) were denatured in NuPAGE LDS sample buffer at 95°C for 5 min and stored at −20°C. The remaining portion of cleared cell extracts was incubated 1 h with protein G-conjugated sepharose beads (GE Healthcare, catalog number 17-0618-01) bound to rabbit anti-GFP antibody under constant rotation for 2–3 h at 4°C. The beads were collected at 4°C by centrifugation at 2000 *g* for 1 min followed by three washings in PBS containing protease inhibitor cocktail (Roche). Immunocomplexes were denatured in NuPAGE LDS sample buffer at 95°C for 5 min followed by SDS-PAGE and WB.

**Northern blotting analysis of ribosomal RNA**. Total RNA (7.5 µg) from human fibroblasts, mouse P19CL6, or zebrafish embryos was separated by denaturing gel electrophoresis on a formaldehyde denaturing 1% agarose gel and subsequently transferred to a positive charged nylon membrane (BrightStar-Plus, Ambion) by capillary blotting, followed by UV cross-linking. The probes (10 pmol each) were labeled with [γ-³²-P]-ATP using T4 polynucleotide kinase (Thermo Fisher Scientific) and hybridized to the membrane in hybridization buffer (4× Denhardts solution, 6× SSC, 0.1% SDS), at 10°C below Tm of the probe for 16 h. The membrane was then washed four times in washing buffer (3× SSC, 0.1% SDS) and exposed to a Phosphor Imager Screen, scanned by a Typhoon scanner (GE Healthcare) and analyzed by ImageQuant TL software version 7.0. The probes used for northern blotting analyses can be found in Supplementary Table 4.

**BrdU assays in HDFs and P19CL6 cells**. A cell proliferation ELISA, BrdU (chemiluminescent) kit (Roche, cat # 11 669 915 001) was used for determining the proliferation status of the cells, according to the manufacturer's instructions. Eighty percent confluent HDFs and P19CL6 cells were trypsinised and resuspended in serum-supplemented growth medium from where 200 µl of ~12,000 cells were seeded in 96-well plates (Cellstar®, cat # 655090) for incubation at 37°C and 5% CO₂ for 24 h. For HDFs, plates were washed in PBS and replaced with 200 µl serum-depleted medium for 48 h to induce growth arrest, followed by re-addition of serum-supplemented medium containing BrdU labeling solution for 18 h. For P19CL6 cells, cells were washed in PBS and incubated in 200 µl serum-enriched medium containing BrdU labeling solution for 4 h. Incubation of HDFs and P19CL6 cells with BrdU was followed by a fixation/denaturing step and incubated with a peroxidase-coupled mouse monoclonal anti-BrdU antibody. The plates were washed extensively, incubated with substrate solution containing luminol, 4-iodophenol and peroxide for 4 min on a shaker, and luminescence was measured using a BMG FLUOstar OPTIMA Microplate Reader. For both HDFs and P19CL6 cultures, cells were counterstained with DAPI to evaluate the levels of BrdU incorporation relative to cell numbers by fluorescence microscopy.

**Zebrafish maintenance**. Embryos were maintained and staged as previously described[71,72] and all experiments were conducted according to the guidelines of the Danish Animal Experiments Inspectorate. The rrp7a (ENSDARG00000098934) mutant line (sa11429) was acquired from Zebrafish International Resource Center (ZIRC)[73]. F3 embryos were raised to adulthood and outcrossed to AB/TL wild-type strain. All analyses were performed on F5-F7 larvae, generated by crosses of heterozygous adult fish.

**Morphological studies of zebrafish larvae**. Survival of the progeny from $rrp7a^{+/-}$ adults was monitored and single larvae were genotyped subsequently. For measurements, lateral and dorsal images of 3 days post fertilization (dpf) zebrafish larvae were taken under Zeiss AxioZoom V16 (Carl Zeiss, Brock Michelsen A/S, Denmark) upon positioning in 3% methylcellulose. Trunk length, eye diameter, and head size (dorsal, lateral) were measured using ImageJ software[74]. The measured absolute values were normalized to the body length of the larvae, which refers to the distance between the tip of the snout and the end of the spine. Head area was defined as outlined (Fig. 8b) by the otic vesicle as posterior and the semicircle of eyes as lower boundary[75]. For sectioning, the 3 dpf zebrafish larvae were collected and fixed in 4% paraformaldehyde overnight at 4 °C. Larvae were washed in 1× PBS, oriented and embedded in 5% agarose. The agarose block was dehydrated in alcohol solutions: 70% EtOH, overnight; 96% EtOH, overnight; 99% EtOH 3×1 h. The block was transferred into Xylene 2×1 h and into paraffin overnight at 60 °C. The block was embedded into fresh paraffin and 2 μm transverse sections were collected. Slides with the sections were dried overnight at 37 °C followed by standard haematoxylin and eosin staining. Stained slides were imaged on a Zeiss Axioplan2 microscope and cell counts were performed using ImageJ Cell Counter[74].

**ISH in zebrafish**. The RT-PCR fragments of *elavl3*, *fgf8a*, *gfap*, *isl1*, *nestin*, *neurod1*, *pcna*, *rrp7a*, and *shha* were amplified using the primers listed in Supplementary Table 4 and were cloned into the pCRII-TOPO vector (Invitrogen) for riboprobe synthesis. The plasmids were linearized and transcribed with SP6 or T7 RNA polymerases (Roche). Digoxygenin (DIG)-labeled probes were used for single ISH and detected with NBT/BCIP (Roche). ISH was performed with minor modifications[76]. In brief, zebrafish larvae were fixed in 4% PFA and in 100% methanol. Larvae were rehydrated, permeabilized using Proteinase K (10 μg ml$^{-1}$), re-fixed in 4% PFA and prehybridized. Larvae were hybridized with RNA probes at 70 °C overnight followed by stringent washes. Larvae were incubated in anti-DIG antibody (1:5000, Roche) and staining was detected with NBT/BCIP (Roche). Larvae were analyzed on a Zeiss Axio Zoom V16 microscope and genotyped subsequently.

**Rrp7a rescue experiments in zebrafish**. Full-length wild-type *rrp7a* (Accession: NM_001017579.1) was amplified from zebrafish cDNA (48 hpf) using primers listed in Supplementary Table 4 and cloned into pCS2 (Addgene) plasmid. Single-nucleotide change that leads to p.W152C was introduced using QuikChange II Site-Directed Mutagenesis Kit (Agilent technologies Inc.), primers listed in Supplementary Table 4. Plasmids with wild-type and mutant *rrp7a* coding sequence and control plasmid pCSGreen (Addgene) were linearized and transcribed using the SP6 mMESSAGE mMACHINE kit (Ambion). 100 pg wild-type *rrp7a*, p. W152C *rrp7a* and GFP mRNA were injected into 1-cell stage embryos from breeding of $rrp7a^{+/-}$ zebrafish. Measurements were performed on images of injected 3 dpf larvae, which were genotyped subsequently.

**qRT-PCR for zebrafish**. Total RNA from pools of 15 zebrafish larvae was extracted using Trizol and used for synthesis of random-primed cDNA (SuperScript II Reverse Transcriptase, Invitrogen). Brilliant III Ultra-Fast SYBR® Green QPCR Master Mix (Agilent Technologies) was used to amplify cDNA, and relative expressions were calculated by normalization against *actb1* and *eef1a1*. Samples were analyzed using a QuantStudio 5 (Applied Biosystems). Primers used are listed in Supplementary Table 4.

**BrdU and TUNEL assays in zebrafish**. BrdU labelling was performed as previously described[77] with slight modifications. In brief, 3 dpf zebrafish larvae were bathed in 10 mM BrdU (ThermoFisher Scientific), 15% dimethyl sulfoxide on ice for 30 min and rinsed with 1× E3 at 28 °C. Thereafter, larvae were fixed in 4% PFA for 2 h and stored in 100% methanol at −20 °C. Larvae were rehydrated, permeabilized using Proteinase K (20 μg ml$^{-1}$), re-fixed in 4% PFA and incubated in 2 N HCl for 1 h at RT. The Vectastain Elite ABC kit (Vectorlabs) was used in the following steps. Larvae were incubated with BrdU antibody (MA1-82718, CiteAb, 1:300) overnight at 4 °C and color was developed using DAB solution (Sigma-Aldrich). For TUNEL staining, dechorionated and fixed larvae were rehydrated and permeabilized with proteinase K (10 μg ml$^{-1}$). ApopTag peroxidase in situ apoptosis detection kit (Millipore) was used for labelling and color was developed using DAB solution (Sigma-Aldrich).

**Morpholino injections**. Antisense morpholinos (MOs) were obtained from GeneTools LLC and 1 nl of MO solution diluted in Danieau's buffer (0.5 pmol) was injected into the yolk of wild-type embryos at the one to two-cell stage. Efficiency of the splice blocking MO (SP-MO) was assessed via RT-PCR on cDNA synthesized from total RNA of SP-MO injected embryos (primers listed in Supplementary Table 4).

**Statistics and reproducibility**. Data are presented as the mean ± SD. The sample size (*n*) indicates the experimental replicates from a single representative experiment; the results of the experiments were validated by independent repetitions. All experiments were carried out with $n \geq 3$, and representative blots and micrographs

are presented. Statistical difference between groups (e.g., WB band and fluorescence intensities) was determined by unpaired, two-sided *t* test or one-way analysis of variance. n.s. indicates not significant, *$p \leq 0.05$, **$p \leq 0.01$, ***$p \leq 0.001$, ****$p \leq 0.0001$. All statistical tests were performed using GraphPad Prism version 8, Sigmaplot version 14.0 or Microsoft Excel version 2010. Statistical analysis of haplotypes was performed using FASTLINK software version 1.0. Logarithm of the odds score was calculated at allele frequency ($p = 0.01$), and disease frequency of 0.001.

**Reporting summary**. Further information on research design is available in the Nature Research Reporting Summary linked to this article.

## Data availability

All relevant data supporting the key findings of this study are available within the article and its Supplementary Information files or from the corresponding author upon reasonable request. Source data, including full scan images of blotting data used in this study, are shown in Source Data file, which is provided with this paper. Exome sequencing data from the single individual, which was analyzed in the family, is not available owing to concerns over patient privacy. Sequence data from the following GenBank accession numbers were used to generate expression constructs: NM_001017579.1 and NM_015703.4. Source data are provided with this paper.

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

## Acknowledgments

We thank Maria S. Holm, Lillian E. Rasmussen, Pernille S. Froh, Troels Askhøj Andersen, and Martin Juel Vilhelm for excellent technical assistance. This study was supported by grants from the Lundbeck Foundation (R221-2016-922), the University of Copenhagen Excellence Program for Interdisciplinary Research, the Novo Nordisk Foundation (NNF18OC0053024, NNF14OC0011535, and NNF15OC0016886), Independent Research Fund Denmark (DFF-6108-00457 and 8020-00162B), the Danish Cancer Society (R146-A9590) and the Carlsberg Foundation (CF15-0425, CF18-0532, and CF-0294) to S.T.C., L.A.L., and L.B.P.; grants from the Lundbeck Foundation (R198-2015-174) and the Danish Cancer Society (R167-A10943-17-S2) to H.N. and N.K.; a grant from the Independent Research Fund Denmark (DNRF107) to L.H.; a grant from The Lundbeck Foundation (R209-2015-2604) to C.D; a grant from the Lundbeck Foundation (2013-14290) to N.T.

## Author contributions

L.A.L., S.T.C., M.F., and L.L. conceived the study. S.T.C. and L.A.L. directed the experiments and wrote the manuscript with input from all authors (M.F., L.L., N.K., C.D., C.K., M.M., A.B.G, S.S., Y.M., A.F., V.S.A., M.S.H., H.E., L.H., K.W.K., J.G., L.B.P., K.M., H.N., S.M.B., and N.T). All authors provided intellectual input to the project. M.F. and S.M.B. identified and performed clinical investigations of the family. M.F., Y.M., H.E., A.F., and L.H. performed genetic analyses. L.L. and K.M. performed immunohistochemistry on foetal human brain sections, which were provided by K.M. L.L., C.K V.S.A., and A.B.G. constructed and analyzed P19CL6 mutant lines. L.L., M.F., and A.B.G. analyzed localization and expression of RRP7A and NOL proteins in HDF and RPE-1 cell lines, and A.B.G performed IP analysis in HEK293T cells. H.N. and N.K. analyzed pre-rRNA processing in HDFs, P19CL6 cells, and zebrafish. L.L. and C.K. analyzed ciliary formation, ciliary resorption, and cell cycle progression in HDFs. C.D., M.M., and S.S. performed zebrafish experiments.

## Competing interests

The authors declare no competing interests.
