## [Peer Review File · Nature Communications]

Reviewers' comments:

Reviewer #1 (Remarks to the Author):

In this manuscript, Farooq et al. are linking primary microcephaly to RRP7A, a gene formerly involved in ribosome biogenesis in eukaryotic cells. The authors are reporting the presence of an homozygous missense mutation in RRP7A in consanguineous individuals with primary microcephaly. The authors are using several experimental models (mouse teratocarcinoma stem cells, patient-derived fibroblasts (HDF), and zebrafish) to build a case for the involvement of RRP7A in neurogenesis, ribosome biogenesis, and ciliary resorption.

This is an interesting work that globally remains quite preliminary. In particular, it is very interesting that RRP7A is localized to the nucleolus, the primary cilia and the centrosome as it suggests very interesting functional connections between ribosome biogenesis and other cell processes. The problem is that we don't really learn much about the nature of such putative connections.

The RRP7A patient mutation appears to mostly affect the localization of the protein at the nucleolus (patient HDF, Fig 3d), and a mild ribosome biogenesis (pre-rRNA processing) defect is reported upon loss of functional RRP7A (Fig 3h), in agreement with a previous report. Note that in HDF cells, pre-rRNA processing is only kinetically delayed and the steady-state amount of mature rRNAs is not reduced. Patient-derived fibroblasts are also defective for primary cilia resorption and cell cycle progression. An important problem is that it remains unclear if it is the ribosome biogenesis defect that is causative of the disease or the role of RRP7A at its other cellular locations (it is not because the protein is still at its other locations that it works there). Other primary microcephaly genes were mapped to the centrosome and cilium biogenesis, so what exactly is the contribution of ribosome biogenesis here? (In my opinion, the sentence in the last paragraph stating "the authors have provided evidence for defective ribosome biogenesis as pathophysiological mechanism" is an overstatement).

Specific comments:

-There is a lack of sufficient background in the Introduction/Discussion (in particular with respect to what is known of the role played by RRP7A in ribosome biogenesis being part of the SSU-processome in close proximity to Utp22).

-The authors have studied only one family and one mutation.

-In the stem cell and zebrafish models, the authors are not characterizing the missense patient mutation (but rather microdeletions, frameshifts etc.).

-The authors have not addressed experimentally in their system if the RRP7A patient mutation does indeed impact the interaction with Utp22, a privileged partner within the SSU-processome. This would have been really nice.

-The use of CRISPR-Cas9 to edit the genome of a mouse cell line that can be differentiated in neurons is interesting but the two isolated cell lines carry compound alleles and microdeletions. Such mutations may have very different effects on ribosome biogenesis by comparison to an amino acid substitution (patient mutation).

From this experiment, it remains unclear if it is a ribosome biogenesis downregulation (not addressed here, but likely) or the presence of a point mutation (as in the patients) in a ribosome assembly factor that is responsible for the effect on neurogenesis. Would the effect on neuronal differentiation be the same with a mutation affecting any other ribosome assembly factors?

-The work on zebrafish is rather preliminary, and would require complementation assays including

injection of rescue mRNAs, and mRNAs carrying the patient mutation. Again, a loss of RRP7A function may or may not mimic the effect of the patient missense mutation.

-Pre-rRNA processing work: both figures 3h and 5i would deserve formal quantification.

Fig 3h: Why is the 18S and 28S RNAs more abundant in the patient and MCPH2 control lanes? Is there a loading issue? It would be useful to control loading. It would also be useful to establish the ratio of mature rRNAs (as an indication of the impact, or not, on ribosomal subunit production).

Fig 5i: What is the molecular nature of the RNA labeled with an asterisk on top of 18S in the rrp7a -/- lane?

Reviewer #2 (Remarks to the Author):

In this paper, Muhammad Farooq et al. report on the identification of RRP7A as a new microcephaly gene. Given the importance of MCPH, especially in relation to neurogenesis, this is a very nice paper that points to novel mechanisms, and it is definitely worth publishing in a prime journal.

I only have a remark concerning data interpretation. Whereas the genetic part of the paper is fully convincing and by far the most important, the cell biological mechanisms proposed are sound but probably still a bit preliminary, and the same can be said of the zebrafish model. The reason for this remark is certainly evident to authors themselves: the mutation W155C is probably not fully recapitulated by null mutations used in cells and zebrafish. W155C could be a null but could also be hypomorph or even dominant-negative (less likely because of recessive inheritance). A mouse or ferret knock-in model would be needed, but this can be done in a follow up paper. Therefore, given the obvious (and understandable) limits of their mechanistic data, I suggest that authors focus their writing and discussion on the strongest part of their data and therefore tune down their title and abstract.

Reviewer #3 (Remarks to the Author):

Farooq et al., linked ribosomal RNA-processing protein 7 homolog A (RRP7A) with primary microcephaly in this study. The strengths of this manuscript come from the human genetic data, reasonable investigations on ribosomal biogenesis and ciliary resorption, and multiple in vitro and in vivo systems used for the mechanistic studies, including cultured cells from patients, CRISPR-generated deletion in mouse progenitor cell model, as well as Zebrafish models. It is exciting to see the link between RRP7A/ribosomal biogenesis and primary microcephaly since the majority studies in the field are focusing on centrosomes and centrosome-related functions including ciliogenesis. Therefore, the potential conceptual advancement of this study to the field is high, given the emerging interests in the heterogeneity and specificity of ribosome biogenesis across a variety of different biology. On the other hand, two major concerns remain for this reviewer. One: is this mutation the cause of microcephaly, neurogenesis, and cilia phenotypes? Patient mutation is ONE missense mutation, which provides a genotype-phenotype correlation and a start place to establish the causative relationship between genotype and phenotype. However, all the CRISPR-mediated mutations in cultured mouse cells and Zebrafish models are frame-shifting mutations at different positions of RRP7A gene, which makes it unclear that whether RRP7A mutation caused microcephaly and if yes, it is loss- or gain-of-function. Second, the ribosomal RNA (rRNA) processing phenotypes are rather subtle, which begs the question that if the ribosomal biogenesis deficit is mediating the neurogenesis and microcephaly phenotype. Alternatively, it could be due to

ciliary resorption that has been linked with primary microcephaly before. Overall, this is a quite interesting study but more work need to be done in order to causatively link the genotype with the phenotype, and functionally link the ribosome biogenesis (not just the name of RRP7A) with primary microcephaly.

Major concerns:

1. Although human missense mutation leads to a decreased RRP7A protein (Fig.S4a), which suggest a loss-of-function mechanism, authors cannot exclude a gain-of-function mechanism and there are still significant residue RRP7A wild type proteins. Therefore, we need to be cautious on phenotypes from the mouse cell line that contain frame-shifting mutations at different position (Fig. 2a) as well as phenotypes from Zebrafish that is a null allele. The concern here is that human mutation and mutation in the functional studies of RRP7A are different in nature. To establish the RRP7A mutation is responsible for the phenotypes in the manuscript, authors should use wild type RRP7A to rescue the phenotypes in patient fibroblast, p19CL6, and Zebrafish. Not all phenotypes need to be rescued, but some of them should be done in order to show that loss of RRP7A is responsible for the deficits. Alternatively, authors should introduce the exactly same mutation into mouse p19CL6 cell line and Zebrafish model for recapitulating the human phenotypes.

2. The rRNA processing phenotypes in patient fibroblast (Fig. 3h) as well as Zebrafish (Fig. 5i) are rather mild without quantification. This begs the question that although RRP7A name is linked with ribosomal biogenesis, it is possible that it is not the impaired ribosomal biogenesis that is responsible for those phenotypes in neurogenesis and microcephaly. Instead, impaired ciliary resorption, which is quite robust based on the data presented, could be the major mediator of the phenotypes. Authors need to convincingly show the ribosomal biogenesis defect, and test if ribosomal biogenesis defect can be rescued by wild type RRP7A proteins.

3. The rRNA processing data in Fig. 3h is not convincing for the following reasons: 1) The difference between control and patient samples are rather subtle. This kind of different is similarly present between control and MCPH2 pt (similar in probe b, and increased accumulation of 30S and 21S in MCPH2 pt comparing to control), which raised the concern that this type of difference is due to ribosomal biogenesis deficit or experimental variation that usually occurs in Northern blot assays; 2) Based on 18S and 28S loading, increased signals in patient and MCPH2 pt could be due to more loading than that in controls. 3) No mention of how many independent experiments were done, and no description of quantification.

4. Fig.5i RNA Northern data is not convincing for the reasons described above.

5. Functional studies in Zebrafish in Fig. 5 were poorly performed. Section immunohistochemical (IHC) staining should be performed to examine cell numbers, cell proliferation, cell death, neural differentiation.

6. Patient data should be described. How do we know the phenotype is autosomal recessive primary microcephaly (MCPH) but not dwarfism or other deficits.

7. Some quantifications are lacking, such as Fig. 2d,e, Fig.3h, Fig.4g p-RB, Fig. 5g, h, i. All statistical methods and n numbers should be clearly presented.

Minor issues:

1. RRP7A expression in cortex is not clear. It is overstated that "showed expression of RRP7A in RGCs at the VZ, ISVZ, OSVZ, intermediate zone/outer fibre layer, subplate and cortical plate as well as localization to ependymal cilia at the VZ (Fig. 1e-h)". Authors need to use apical neuro-progenitor marker (Pax6), intermediate progenitor marker Tbr2, layer markers (such as *cux1*, *Satb2*, *Ctip2*) coupled with dorsal-ventral axis of cortex (such as DAPI staining) to define the different brain regions followed by RRP7A staining. OSVZ can be marked by Sox2+Pax6- staining. In addition, individual zoom in images should be provided for its localization in cilia, nucleoli, as

well as cytoplasm in the neural progenitor cells in the developing cortex in Figure 1.

2. Authors should use BrdU labeling, immunohistochemical (IHC) staining (Ki67, p-H3) to assess proliferation in addition to cell growth in Fig. 2b,c.

3. Fig. 2d,e need to be quantified.

4. Pax6 protein level should be quantified in Fig. 2f.

5. RPP7A is required for timely neurogenesis, but "onset of neurogenesis" is an overinterpretation. Also, "formation of elaborate networks of post-mitotic neurons" is overstated in the first paragraph of Page 5.

6. The logic is wrong that "ribosome biogenesis in RRP7A mutant cells is correlated with a delay in cell cycle progression", therefore the authors test this by studying cilium in second paragraph of page 6.

7. Authors should use BrdU labeling to examine cell cycle re-entry since delayed cilium assembly should disrupt cell cycle re-entry. Although p-RB is assessed in Fig. 4g,h, the standard assay of investigating the effect of ciliogenesis defect on cell cycle is the cell cycle re-entry assay, which can be easily performed in cultured cells.

8. Western blot should be performed to examine RRP7A protein in mutant Zebrafish. So far, there is only a RT-PCR analysis of mRNA in Fig. S5b.

Point-by point response to the reviewer's comments

Reviewer #1.

In this manuscript, Farooq et al. are linking primary microcephaly to RRP7A, a gene formerly involved in ribosome biogenesis in eukaryotic cells. The authors are reporting the presence of an homozygous missense mutation in RRP7A in consanguineous individuals with primary microcephaly. The authors are using several experimental models (mouse teratocarcinoma stem cells, patient-derived fibroblasts (HDF), and zebrafish) to build a case for the involvement of RRP7A in neurogenesis, ribosome biogenesis, and ciliary resorption.

This is an interesting work that globally remains quite preliminary. In particular, it is very interesting that RRP7A is localized to the nucleolus, the primary cilia and the centrosome as it suggests very interesting functional connections between ribosome biogenesis and other cell processes. The problem is that we don't really learn much about the nature of such putative connections.

The RRP7A patient mutation appears to mostly affect the localization of the protein at the nucleolus (patient HDF, Fig 3d), and a mild ribosome biogenesis (pre-rRNA processing) defect is reported upon loss of functional RRP7A (Fig 3h), in agreement with a previous report. Note that in HDF cells, pre-rRNA processing is only kinetically delayed and the steady-state amount of mature rRNAs is not reduced. Patient-derived fibroblasts are also defective for primary cilia resorption and cell cycle progression. An important problem is that it remains unclear if it is the ribosome biogenesis defect that is causative of the disease or the role of RRP7A at its other cellular locations (it is not because the protein is still at its other locations that it works there). Other primary microcephaly genes were mapped to the centrosome and cilium biogenesis, so what exactly is the contribution of ribosome biogenesis here? (In my opinion, the sentence in the last paragraph stating "the authors have provided evidence for defective ribosome biogenesis as pathophysiological mechanism" is an overstatement).

We thank the reviewer for the very constructive comments. The reviewer raises some important queries, which we have now addressed in writing and by including new data.

Specific comments:

1) There is a lack of sufficient background in the Introduction/Discussion (in particular with respect to what is known of the role played by RRP7A in ribosome biogenesis being part of the SSU-processome in close proximity to Utp22).

We agree and increased background information and discussion on this are now included in the revised manuscript.

2) The authors have studied only one family and one mutation.

We agree that it would have been more powerful to identify RRP7A mutations in more than one MCPH family and we have submitted RRP7A as a microcephaly gene to GeneMatcher in Aug. 2017. We have not received any responses from the community on the gene, suggesting that our family is rare. However, we would like to emphasize that our MCPH locus segregates with MCPH in a very large family (with LOD score Z=8.61).

3) In the stem cell and zebrafish models, the authors are not characterizing the missense patient mutation (but rather microdeletions, frameshifts etc.).

To comply with this in zebrafish, we now provide more detailed information on the molecular and cellular processes underlying the microcephaly phenotype in *rrp7a*^{-/-} fish, and we demonstrate that the microcephaly phenotype can be rescued with wt *rrp7a* mRNA, but not with *rrp7a* W152C mutant mRNA, supporting the

conclusion that the phenotype-genotype relationship observed in the family can be replicated in the zebrafish model. The new data is shown in Figure 6h and Supplementary Figure 9. Similar rescue experiments were also attempted in P19CL6 stem cells, but due to technical difficulties in variability in transfection and expression rates of WT and mutant constructs, we decided to focus our efforts on the zebrafish.

4) The authors have not addressed experimentally in their system if the RRP7A patient mutation does indeed impact the interaction with Utp22, a privileged partner within the SSU-processome. This would have been really nice.

We have now included data based on co-IP experiments, showing that the p.W155C mutation in RRP7A produces a prominent reduction in the interaction with NOL6 (UTP22), and that this is correlated with reduced nucleolar localization of NOL6. In contrast, interaction between RRP7A and Nucleolar Protein 12 (NOL12) is not compromised by the p.W155C mutation and nucleolar localization of NOL12 is not affected (Figure 4h-j). These results are in line with the idea that defect in formation of 18S rRNA in patient HDFs is coupled to reduced interaction between RRP7A and NOL6, thereby compromising nucleolar recruitment of a fully functional RRP7A-NOL6-containing SSU processome complex.

5) The use of CRISPR-Cas9 to edit the genome of a mouse cell line that can be differentiated in neurons is interesting but the two isolated cell lines carry compound alleles and microdeletions. Such mutations may have very different effects on ribosome biogenesis by comparison to an amino acid substitution (patient mutation). From this experiment, it remains unclear if it is a ribosome biogenesis downregulation (not addressed here, but likely) or the presence of a point mutation (as in the patients) in a ribosome assembly factor that is responsible for the effect on neurogenesis. Would the effect on neuronal differentiation be the same with a mutation affecting any other ribosome assembly factors?

*We acknowledge this comment, but due to technical difficulties with variability in transfection and expression rates of WT and mutant constructs in P19CL6 mutant cells, we decided to focus our efforts on the zebrafish. However, we now provide information on the P19CL6 mutations, which result in deletion within the N-terminal domain (NTD) that makes extensive interactions with Utp22 based on X-ray crystallography of the yeast complex (Lin et al., 2013), and we show that these mutants display a ribosome biogenesis phenotype (Supplemental Figure 5a) comparable to that of patient cells and *rrp7a*^{-/-} fish. Further, we show that nucleolar localization of NOL6 (and not NOL12) is reduced in P19CL6 mutant cells in a manner identical to that observed in patient HDFs (Supplemental Figure 5b-d).*

6) The work on zebrafish is rather preliminary, and would require complementation assays including injection of rescue mRNAs, and mRNAs carrying the patient mutation. Again, a loss of RRP7A function may or may not mimic the effect of the patient missense mutation.

*We have performed rescue experiments with wt and mutant *rrp7a* mRNA (see above). In addition, we have injected fertilized oocytes with two different morpholino oligonucleotides, targeted at *rrp7a*. The phenotype of both type of morphants are similar to the *rrp7a* mutants (Supplementary figure 6e-g). Furthermore, we have performed analysis of additional markers, and performed proliferation and apoptosis assays (Supplementary Figure 7 and 8).*

7) Pre-rRNA processing work: both figures 3h and 5i would deserve formal quantification.

Quantification of the autoradiograms has been added to the figure on patient RNA (now Fig. 4g). In the zebrafish model, we have made new northern blot experiments using ITS-probes based on a recent paper (Locati et al., RNA 23:1188–1199 (17)) instead of probes targeting the mature species (now Fig. 6i). These northern blots more clearly visualize processing intermediates. In addition, we have performed RT-qPCR experiments to demonstrate clearly the decreased 18S/28S rRNA ratio.

8) Fig 3h: Why is the 18S and 28S RNAs more abundant in the patient and MCPH2 control lanes? Is there a loading issue? It would be useful to control loading. It would also be useful to establish the ratio of mature rRNAs (as an indication of the impact, or not, on ribosomal subunit production).

Indeed there is a loading issue. However, the idea of the figure was to demonstrate altered levels of processing intermediates which does not critically depend on the amount loaded. The ratio of the mature rRNA did not deviate significantly from 1:1 stoichiometry. With respect to cellular levels of rRNA, a reliable cell count was not obtained to make any meaningful statement on this.

9) Fig 5i: What is the molecular nature of the RNA labeled with an asterisk on top of 18S in the *rrp7a*^{-/-} lane?

*The pre-rRNA processing intermediates in zebrafish have not been mapped to the same detail as in mouse and human. However, we have re-probed the filter with several additional probes and conclude that the band labeled with an asterisk in the previous version of the manuscript is a normal processing intermediate (corresponding to 30S in humans) that was detected in the *rrp7*^{-/-} lane because these fish have much higher levels of intermediates than the other samples.*

Reviewer #2.

In this paper, Muhammad Farooq et al. report on the identification of RRP7A as a new microcephaly gene. Given the importance of MCPH, especially in relation to neurogenesis, this is a very nice paper that points to novel mechanisms, and it is definitely worth publishing in a prime journal.

I only have a remark concerning data interpretation. Whereas the genetic part of the paper is fully convincing and by far the most important, the cell biological mechanisms proposed are sound but probably still a bit preliminary, and the same can be said of the zebrafish model. The reason for this remark is certainly evident to authors themselves: the mutation W155C is probably not fully recapitulated by null mutations used in cells and zebrafish. W155C could be a null but could also be hypomorph or even dominant-negative (less likely because of recessive inheritance). A mouse or ferret knock-in model would be needed, but this can be done in a follow up paper. Therefore, given the obvious (and understandable) limits of their mechanistic data, I suggest that authors focus their writing and discussion on the strongest part of their data and therefore tune down their title and abstract.

We thank the reviewer for the very enthusiastic reaction to our manuscript and suggestions for improvements. We have addressed the issue concerning differences in mutation type between patients and the zebrafish model by in vivo rescue experiments (data shown in Figure 6h and Supplementary figure 9). We demonstrate that the microcephaly phenotype in zebrafish is rescued by injection of wild-type (but not mutant W152C) rrp7a mRNA into progeny from rrp7^{+/-} fish, supporting that the phenotype is specific to mutation of rrp7a and importantly, that the phenotype-genotype relationship observed in the Pakistani family is replicated in the zebrafish model. Similar rescue experiments were also attempted in P19CL6 stem cells, but due to technical difficulties with variability in transfection and expression rates of WT and mutant constructs, we decided to focus our efforts on the zebrafish. Further, we demonstrate that the p.W155C mutation in RRP7A produces a prominent reduction in the interaction with NOL6 (UTP22), and that this is correlated with reduced nucleolar localization of NOL6. In contrast, the interaction between RRP7A and Nucleolar Protein 12 (NOL12) is not compromised by the p.W155C mutation and nucleolar localization of NOL12 is not affected. Similar results were obtained with P19CL6 stem cell mutants, which have deletions within the N-terminal domain (NTD) that makes extensive interactions with Utp22 based on X-ray crystallography of the yeast complex (Lin et al., 2013), supporting the conclusion that defect in formation of 18S rRNA in patient HDFs as well as P19CL6 stem cell mutants is coupled to reduced interaction between RRP7A and NOL6, thereby compromising nucleolar recruitment of a fully functional RRP7A-NOL6-containing SSU processome complex. Please see our response to Reviewer #1, who raised the same issue, above.

Reviewer #3.

Farooq et al., linked ribosomal RNA-processing protein 7 homolog A (RRP7A) with primary microcephaly in this study. The strengths of this manuscript come from the human genetic data, reasonable investigations on ribosomal biogenesis and ciliary resorption, and multiple in vitro and in vivo systems used for the mechanistic studies, including cultured cells from patients, CRISPR-generated deletion in mouse progenitor cell model, as well as Zebrafish models. It is exciting to see the link between RRP7A/ribosomal biogenesis and primary microcephaly since the majority studies in the field are focusing on centrosomes and centrosome-related functions including ciliogenesis. Therefore, the potential conceptual advancement of this study to the field is high, given the emerging interests in the heterogeneity and specificity of ribosome biogenesis across a variety of different biology. On the other hand, two major concerns remain for this reviewer. One: is this mutation the cause of microcephaly, neurogenesis, and cilia phenotypes? Patient mutation is ONE missense mutation, which provides a genotype-phenotype correlation and a start place to establish the causative relationship between genotype and phenotype. However, all the CRISPR-mediated mutations in cultured mouse cells and Zebrafish models are frame-shifting mutations at different positions of RRP7A gene, which makes it unclear that whether RRP7A mutation caused microcephaly and if yes, it is loss- or gain-of-function. Second, the ribosomal RNA (rRNA) processing phenotypes are rather subtle, which begs the question that if the ribosomal biogenesis deficit is mediating the neurogenesis and microcephaly phenotype. Alternatively, it could be due to ciliary resorption that has been linked with primary microcephaly before. Overall, this is a quite interesting study but more work need to be done in order to causatively link the genotype with the phenotype, and functionally link the ribosome biogenesis (not just the name of RRP7A) with primary microcephaly.

We thank the reviewer for the very constructive comments. The reviewer raises some important queries, which we have now addressed in writing and by including new data.

Major concerns:

1) Although human missense mutation leads to a decreased RRP7A protein (Fig.S4a), which suggest a loss-of-function mechanism, authors cannot exclude a gain-of-function mechanism and there are still significant residue RRP7A wild type proteins. Therefore, we need to be cautious on phenotypes from the mouse cell line that contain frame-shifting mutations at different position (Fig. 2a) as well as phenotypes from Zebrafish that is a null allele. The concern here is that human mutation and mutation in the functional studies of RRP7A are different in nature. To establish the RRP7A mutation is responsible for the phenotypes in the manuscript, authors should use wild type RRP7A to rescue the phenotypes in patient fibroblast, p19CL6, and Zebrafish. Not all phenotypes need to be rescued, but some of them should be done in order to show that loss of RRP7A is responsible for the deficits. Alternatively, authors should introduce the exactly same mutation into mouse p19CL6 cell line and Zebrafish model for recapitulating the human phenotypes.

We have addressed the issue concerning differences in mutation type between patients and our model by in vivo rescue experiments in zebrafish. Please see our response to Reviewer #1, who raised the same issue, above.

2) The rRNA processing phenotypes in patient fibroblast (Fig. 3h) as well as Zebrafish (Fig. 5i) are rather mild without quantification. This begs the question that although RRP7A name is linked with ribosomal biogenesis, it is possible that it is not the impaired ribosomal biogenesis that is responsible for those phenotypes in neurogenesis and microcephaly. Instead, impaired ciliary resorption, which is quite robust based on the data presented, could be the major mediator of the phenotypes. Authors need to convincingly show the ribosomal biogenesis defect, and test if ribosomal biogenesis defect can be rescued by wild type RRP7A proteins.

Quantification of the autoradiogram (patient analysis; now Fig. 4g) and a supplementary RT-qPCR experiment (zebrafish; now fig. 6i) have been added to the two figures. In addition, we have demonstrated that the morphological phenotype in zebrafish could be rescued by injection of WT, but not p.W155C mRNA. Unfortunately, it was not technically feasible to extend the rescue analysis to the pre-rRNA processing phenotype because this would require genotyping, pre-rRNA processing analysis, and validation of productive injection of mRNA in each individual progeny.

We do not find the pre-rRNA processing phenotype to be subtle considering that the mutation was characterized from adult patients. In fact, we find the effect to be comparable to that reported from patient cells isolated from Diamond-Blackfan patients (e.g. as reported in Doherty et al., AJHG 86:222-228 (10)). In the present study, we attempted to exacerbate the effect of RRP7A mutations in the model systems. In doing so, it was clearly demonstrated by the inability to make CRISPR-KO cell lines in the mouse and by lethality of the KO in zebrafish that RRP7A is an essential gene. Thus, it is likely that mutations that severely impact ribosome biogenesis will be selected against during early development.

With respect to the causality, we agree that more work is needed to conclude that the pre-rRNA processing defect is solely responsible for the microcephaly phenotype, although the rescue experiment in the zebrafish provided important additional support. One of the main unknowns is obviously to what extent studies of patient fibroblasts reflect the manifestation of the mutation during neurogenesis. It is possible; perhaps even likely, that neural development is more sensitive to perturbations of ribosome biogenesis. It is also possible that the RRP7A mutation in addition to the effect on pre-rRNA processing, impacts other aspects of ribosome biogenesis, e.g. ribosomal protein genes that are in part regulated by the CURI complex.

With respect to the additional effects observed in RRP7A mutants in the present study, we note that in siRNA depletion of RRP7A in human cell lines (Tafforeau et al.), the cell cycle arrest and apoptosis was downstream of the pre-rRNA processing defect. An interesting possibility is that the defect in NOL-localization results in increased availability of RRP7A for functions in cilia or centrosomes, but this must await future work on elucidating the role of RRP7A at these locations.

3)The rRNA processing data in Fig. 3h is not convincing for the following reasons: 1) The difference between control and patient samples are rather subtle. This kind of difference is similarly present between control and MCPH2 pt (similar in probe b, and increased accumulation of 30S and 21S in MCPH2 pt comparing to control), which raised the concern that this type of difference is due to ribosomal biogenesis deficit or experimental variation that usually occurs in Northern blot assays; 2) Based on 18S and 28S loading, increased signals in patient and MCPH2 pt could be due to more loading than that in controls. 3) No mention of how many independent experiments were done, and no description of quantification.

We have performed similar pre-rRNA processing analyses at numerous occasions and have not observed similar changes due to experimental variation. We agree that the northern looked less convincing prior to the inclusion of quantification due to less RNA loaded in the control lane, but hope that it now is apparent that the patient, and not the MCPH2 sample, differs from the control, as described.

4)Fig.5i RNA Northern data is not convincing for the reasons described above.

The northern blot experiment has been supplemented with an RT-qPCR experiment that clearly demonstrates the decreased 18S/28S rRNA ratio.

5) Functional studies in Zebrafish in Fig. 5 were poorly performed. Section immunohistochemical (IHC) staining should be performed to examine cell numbers, cell proliferation, cell death, neural differentiation. *We have quantified the loss of cell numbers in the pallium by image analyses (Figure 6g, right panel). We have analyzed effects on cell proliferation by BrdU staining and analysis of proliferation marker pcna, by WISH and quantitative RT-PCR (Supplementary Figure 7a). Cell death has been examined by TUNEL assay and analysis of*

apoptosis markers (Supplementary Figure 8) and neural differentiation was analyzed by WISH and quantitative RT-PCR analysis of neural differentiation markers (Supplementary Figure 7b).

6) Patient data should be described. How do we know the phenotype is autosomal recessive primary microcephaly (MCPH) but not dwarfism or other deficits.

The clinical phenotype of eight affected family members is listed in Supplementary table 1. The clinical phenotype of the patients fits with the clinical definition of MCPH.

7) Some quantifications are lacking, such as Fig. 2d,e, Fig.3h, Fig.4g p-RB, Fig. 5g, h, i. All statistical methods and n numbers should be clearly presented.

All data presented in the figures are now presented with quantifications, and statistical methods and n numbers are included in the legends.

Minor issues:

1) RRP7A expression in cortex is not clear. It is overstated that “showed expression of RRP7A in RGCs at the VZ, ISVZ, OSVZ, intermediate zone/outer fibre layer, subplate and cortical plate as well as localization to ependymal cilia at the VZ (Fig. 1e-h)”. Authors need to use apical neuro-progenitor marker (Pax6), intermediate progenitor marker Tbr2, layer markers (such as *cux1*, *Satb2*, *Ctip2*) coupled with dorsal-ventral axis of cortex (such as DAPI staining) to define the different brain regions followed by RRP7A staining. OSVZ can be marked by Sox2+Pax6- staining. In addition, individual zoom in images should be provided for its localization in cilia, nucleoli, as well as cytoplasm in the neural progenitor cells in the developing cortex in Figure 1.

We agree that the immunoreactivity of RRP7A and in particular that in radial glial fibers may not be followed through the cortical wall in the low magnification pictures provided. In the revised manuscript, we have added larger magnification of the images and we have rephrased the sentence to “showed expression of RRP7A in RGCs in VZ and SVZ and in RG fibers in the cortical plate and marginal zone as well as localization to ependymal cilia at the VZ (Fig. 2a-k)”. Following the three original articles published in 2002 (Smart et al., Altman & Bayer, Kostovic et al.) the laminar zones in human (and primate) mid-gestation cortical wall are by now well-characterized by ordinary histology based on Nissl-stained sections (see very recent paper by Kostovic et al., 2019). We have added this reference after the description of the individual zones shown in Supplementary Fig. 2a stained for the RGC marker par excellence Vimentin. Immunostainings were further evaluated for localization of RRP7A to primary cilia and nucleoli in neural progenitor cells, but the material was not of sufficient quality to address subcellular localizations of RRP7A. Future studies will address subcellular localizations in the developing human brain by obtaining more of this rare material.

2) Authors should use BrdU labeling, immunohistochemical (IHC) staining (Ki67, p-H3) to assess proliferation in addition to cell growth in Fig. 2b,c.

We have now included BrdU labeling, which confirms that proliferation is reduced in mutant cell lines (Figure 3d).

3) Fig. 2d,e need to be quantified.

Data presented for day 9 are now presented with quantifications (Figure 3g).

4) Pax6 protein level should be quantified in Fig. 2f.

These data are now presented with quantifications (Figure 3i).

5) RRP7A is required for timely neurogenesis, but “onset of neurogenesis” is an overinterpretation. Also, “formation of elaborate networks of post-mitotic neurons” is overstated in the first paragraph of Page 5.

We agree and have rephrased accordingly

6) The logic is wrong that “ribosome biogenesis in RRP7A mutant cells is correlated with a delay in cell cycle progression”, therefore the authors test this by studying cilium in second paragraph of page 6.

We agree and have rephrased accordingly

7) Authors should use BrdU labeling to examine cell cycle re-entry since delayed cilium assembly should disrupt cell cycle re-entry. Although p-RB is assessed in Fig. 4g,h, the standard assay of investigating the effect of ciliogenesis defect on cell cycle is the cell cycle re-entry assay, which can be easily performed in cultured cells.

We have now included BrdU labeling, which confirms that proliferation by S-phase entry/progression is hampered in patient cells (Figure 5j). Further, we have included western blotting analysis showing that patient HDFs during serum re-introduction display delayed phosphorylation of CDK1 at threonine in position 161 (Figure 5i), which marks S/G2 phase transition.

8) Western blot should be performed to examine RRP7A protein in mutant Zebrafish. So far, there is only a RT-PCR analysis of mRNA in Fig. S5b.

We have tried to perform WB using available antibodies, but unfortunately, they do not cross-react with zebrafish.

Reviewers' comments:

Reviewer #1 (Remarks to the Author):

This is quite a nice study. The connection with primary cilium (resorption, second wave) is very clear. Unfortunately, much less is the connection with ribosome biogenesis (nucleolar localization apart). I completely agree with referee #3 on this: the effect on primary cilium resorption may be only effect that matters.

I appreciate that rescue analysis were performed in zebrafish, and that the effect on UTP component interaction was tested.

Nonetheless, the pre-rRNA processing analysis should still be strengthened to make a case on a role of ribosome biogenesis in this disease.(see also referee #3 on this).

The processing data in patient(human)-derived dermal fibroblasts (HDFs) should be triplicated, as some phenotypes remain quite difficult to see or simply do not agree with the authors' conclusions (loss of 18S). If there is also a 'loading issue' (rebuttal, #1 point 8), then the gel should be reloaded.

The processing data in fish is also not easy to understand.

Specific comments:

1) Fig 4g, patient cells processing analysis:

The loss of 41S in the patient is rather weak and difficult to see (Fig 4g)

The accumulation of 30S is easier to see

The depletion of 21S, I simply don't see.

In their discussion, the authors conclude that the patient cells produce less 18S rRNA, but this is really not obvious from Fig 4g lower panels.

2) Fig 6i, fish processing data

How is the loss of 18S explained in terms of processing inhibitions?

There seem to be more of the precursors leading to 18S rRNA in the rrp7a mutant.

Unclear.

Reviewer #2 (Remarks to the Author):

Authors have done their best to answer issues raised in reviews. Although additional data do not prove every aspect of the model proposed for them mutation's effects, the paper is very significantly improved. I was already positive about the first version and the present manuscript is clearly improved and worth publishing as such, at least as far as I am concerned.

Reviewer #3 (Remarks to the Author):

The authors were responsive and added a significant amount of new data. The manuscript is significantly improved.

Point-by point response to the reviewer's comments

We wish to thank the reviewers for constructive criticism, which in our mind has improved the manuscript. As requested by reviewer #1, the northern blot analysis of rRNA processing in patient fibroblasts has now been performed in triplicates. Thus, we believe to have responded to all points raised by this reviewer. Please find our response below the specific comments.

Reviewer #1:

This is quite a nice study. The connection with primary cilium (resorption, second wave) is very clear. Unfortunately, much less is the connection with ribosome biogenesis (nucleolar localization apart). I completely agree with referee #3 on this: the effect on primary cilium resorption may be only effect that matters. I appreciate that rescue analysis were performed in zebrafish, and that the effect on UTP component interaction was tested. Nonetheless, the pre-rRNA processing analysis should still be strengthened to make a case on a role of ribosome biogenesis in this disease (see also referee #3 on this). The processing data in patient(human)-derived dermal fibroblasts (HDFs) should be triplicated, as some phenotypes remain quite difficult to see or simply do not agree with the authors' conclusions (loss of 18S). If there is also a 'loading issue' (rebuttal, #1 point 8), then the gel should be reloaded.

Response from authors: *The loss of 18S is only observed in the zebrafish model. To clarify this and to strengthen the argument of a role of the defect in rRNA processing as part of the pathophysiological mechanism, we have slightly rephrased the first and second section on page 13 and added the two following sentences:*

1) *"Patient HDFs, P19CL6 cells, and zebrafish mutants all displayed an rRNA processing phenotype specifically affecting production of 18S rRNA. The processing defects scaled with the genetic insults. In the cell lines, that suffered point mutations (patient HDFs) or compound heterozygous frameshift and in-frame deletion mutations (P19CL6) we observed changes of specific processing intermediate levels. In the *rrp7a*^{-/-} zebrafish, the effect extended to depletion of 18S rRNA and thus an imbalance between mature rRNA species. The consistent observation of processing defects affecting 18S rRNA in the three model systems supports that defective rRNA processing is part of the pathophysiological mechanism."*

2) *"However, even a milder kinetic effect on ribosome biogenesis, as observed in the cell line models, may impact ribosome production and cell growth. Indeed, the patient HDFs grow considerable slower than WT cells. Furthermore, evidence is accumulating that changes in ribosome dose can dramatically affect the cellular proteome and cell fate (Pelletier et al. Nat. Rev. Cancer 18:51-63 (2017), Ferretti and Karbstein RNA 25: 521-538 (2019)). Thus, in addition to similar effects on rRNA processing, the growth phenotypes in the three models appear related, with slow growth in HDFs, defects in neuronal differentiation in P19CL6 stem cells and reduced brain size in zebrafish. Taken together, the three model systems that were subjected to different genetic insults of *rrp7a* show corresponding defects in rRNA processing and growth and differentiation phenotypes supporting the interpretation that one consequence of the p.W155C patient mutation is on rRNA production, although further work will be required to delineate the precise consequences of ribosome insufficiency on brain development such as in proliferation-differentiation decisions of RGCs in the developing neocortex."*

To strengthen the rRNA processing defects (accumulation/reduction of intermediates) seen in patient HDFs compared to control and WDR62 patient fibroblasts, respectively, the northern blots were performed in triplicates as suggested by the reviewer (please see the updated quantification in Fig. 4g and the additional northern blots below; answers related to the northern blots are added to each specific comments raised below).

The processing data in fish are also not easy to understand. rRNA processing in zebrafish is currently poorly understood compared to the two other models (human and mice). To emphasize this, we have slightly rephrased the last section on page 11 and added the following sentence: "Zebrafish pre-rRNA processing has not been studied in sufficient detail to link the observed accumulation of intermediates to the depletion of 18S rRNA. Also, other possibilities, e.g. increased transcription of rRNA genes in the mutant, cannot be ruled out at present."

Specific comments:

1) Fig 4g, patient cells processing analysis: The loss of 41S in the patient is rather weak and difficult to see (Fig 4g)

We agree that 41S is faint on the northern blots, but this is consistent with low abundance of this intermediate. Nevertheless, the reduction of 41S in the patient is still evident after performing the northern blots in triplicates. We hope this is clearer now.

The accumulation of 30S is easier to see

This is still evident after performing the northern blots in triplicates.

The depletion of 21S, I simply don't see.

The depletion of 21S relative to 45S is hopefully seen more clearly after the northern blots are triplicated.

In their discussion, the authors conclude that the patient cells produce less 18S rRNA, but this is really not obvious from Fig 4g lower panels.

We apologize for this misunderstanding and hope to have clarified in the discussion section that reduction of 18S rRNA is only seen in the model system (zebrafish) with the most severe genetic insult of rrp7a. Please see above.

2) Fig 6i, fish processing data. How is the loss of 18S explained in terms of processing inhibitions? There seem to be more of the precursors leading to 18S rRNA in the rrp7a mutant. Unclear.

We agree with this, in fact there seems to be more of the intermediates leading to both 18S and 28S, respectively. There could be several reasons for this: 1) rDNA transcription could be up-regulated to compensate for loss of 18S 2) There could be a processing defect similar to the observations in patient HDFs and mouse P19CL6 cells that escape our attention due the incomplete mapping of pre-rRNA processing in zebrafish. However, we find that elaborating on this would become too speculative at present.

Reviewer #2:

Authors have done their best to answer issues raised in reviews. Although additional data do not prove every aspect of the model proposed for them mutation's effects, the paper is very significantly improved. I was already positive about the first version and the present manuscript is clearly improved and worth publishing as such, at least as far as I am concerned.

Reviewer #3:

The authors were responsive and added a significant amount of new data. The manuscript is significantly improved.

REVIEWERS' COMMENTS:

Reviewer #1 (Remarks to the Author):

The authors have adequately addressed my comments, and I would like to congratulate them for a very interesting contribution.

Denis L.J. Lafontaine